# DeFa: Non-Stationary Decomposition and Factorized Forecasting for Multivariate Time Series

## Abstract

Multivariate time series forecasting is essential in fields like energy systems, weather prediction, and traffic monitoring. While recent deep learning models, including Transformer-based architectures, show potential, they often struggle to capture the complex dynamics and non-stationary patterns inherent in real-world data. This limitation arises from over-parametrization and the difficulty in modelling shifting patterns in simple short- and long-term terms. In this paper, we propose a unified framework, **DeFa**, that addresses these challenges by combining decomposition-based modelling with tensor autoregressive forecasting. To capture long-term dynamics, stationary seasonality, and sparse residuals unique to non-stationary time series, DeFa decomposes the input series into three components using the **N**on-stationary **A**daptive **I**nteractive **Long**-term strategy (**NAILong**). Furthermore, to improve the prediction of the Amplifier, which encodes time-varying dynamics, DeFa is enhanced with the **Fa**ctorized **T**ensor **A**utoregression framework (**FaTA**). Unlike existing methods that disentangle or represent input series directly, FaTA explicitly models the autoregressive coefficient tensor across variates and temporal dimensions. This fusion enables a more flexible and interpretable representation of multi-variable interactions, improving forecasting accuracy while maintaining computational efficiency. Extensive experiments on real-world datasets show that DeFa outperforms state-of-the-art methods in terms of both interpretable forecasting accuracy and scalability. Additionally, DeFa handles long-term dynamics and drifting seasonalities efficiently through a plug-in option, extending its adaptability.

## 1 Introduction

Multivariate time series forecasting plays a pivotal role in diverse fields such as energy systems, weather prediction, and traffic monitoring Hao & Liu (2024); Papastefanopoulos et al. (2023). These real-world datasets are inherently complex, exhibiting non-stationary dynamics, multi-scale temporal dependencies, and cross-variable interactions. Accurate long-term forecasting must address these complexities while maintaining both interpretability and computational efficiency.

Recent deep learning approaches to multivariate time series forecasting have adopted three primary paradigms. Lightweight linear layers assume stationarity, focusing on simple affine transformations Lai et al. (2018); Liu et al. (2022); Lin et al. (2024b); Li et al. (2019); Zhang et al. (2022); Xu et al. (2024). Transformer-based methods and large language models (LLMs) treat time series as trend and seasonal decompositions during training Li et al. (2023b); Zhang & Yan (2023), enabling the representation of inter-channel dynamics through self-attention Wen et al. (2022); Zhou et al. (2021); Wu et al. (2021); Liu et al. (2021); Zhou et al. (2022); Zhang & Yan (2023); Liu et al. (2024b). Additionally, some approaches align temporal patterns with linguistic structures Nie et al. (2023); Wang et al. (2024b) or integrate time prompts into LLMs Wu et al. (2023); Liu et al. (2024c); Jin et al. (2023); Gruver et al. (2024); Liu et al. (2024a). Despite their initial promise, these models often produce forecasts that resemble random noise over long time horizons Huang et al. (2024); Su et al. (2025), with recent studies indicating that they often underperform compared to simple linear baselines Tian et al. (2023); Bergmeir (2024); Zeng et al. (2023); Chen et al. (2025). In addition, frequency-domain methods, such as spectral filters (Fourier/Wavelet) and hybrid time-

frequency representations, also struggle to address the dynamic nature of real-world data Dong et al. (2024); Wang et al. (2025); Zhou et al. (2022); Piao et al. (2024); Yue et al. (2025); Wu et al. (2025). None of these models—whether based on attention maps in transformers, spectral coefficients in frequency-domain methods, or latent projections in linear models—offer insights grounded in real-world physics Tian et al. (2023); Bergmeir (2024); Zeng et al. (2023); Chen et al. (2025). Linear latent spaces fail to interpret causal time-shifting, such as holiday effects Ilbert et al. (2024b). Attention weights in transformers indicate correlations between variables but do not capture the directional dependencies dictated by physical laws within the dataset Wu et al. (2023); Liu et al. (2024c); Jin et al. (2023); Gruver et al. (2024); Liu et al. (2024a). Similarly, spectral methods treat patterns as short-term periods or long-term seasonality but may obscure critical local events due to the limitations of spectral resolution Fu & Hu (2025).

The primary challenge in multivariate forecasting lies in effectively identifying and modeling time-varying behaviors. Existing frameworks rely on additive formulations to verify stationarity, separating short-term and long-term seasonalities through global normalizations. However, these additive methods provide limited representations, particularly when dealing with time-varying dynamics that span multiple time scales. They overlook the crucial adaptability required to model non-stationary patterns, such as interactions between multi-scale trends and seasonality.

To address these issues, we propose **DeFa**, a decomposition-and-forecast framework for non-stationary time series forecasting that first decomposes historical series, then forecasts future series. DeFa is first grounded in a novel decomposition framework called Non-Stationary Adaptive Interaction (**NAILong**). Unlike additive paradigms that conflate components, NAILong decomposes non-stationary time series into three key matrixes: a time-varying amplifier $Amp$, a stationary normalized seasonality $NS$, and a sparse residual $R$. Formally, NAILong expresses the series as $X = Amp \cdot NS + R$, holding different component definitions than the stationary linear representation. The amplifier **Amp** encodes unidirectional temporal data and multi-channel interactions. The normalized seasonality **NS** preserves stationary patterns, while the residual **R** captures anomalies or irregular events.

To forecast the long-standing multi-scale shifts and pattern drifts Zhou & Yu (2025); Stitsyuk & Choi (2025) in the dynamics of $Amp$, we introduce the Factorised Tensor Autoregressive (**FaTA**) module, which applies tensor decomposition to the autoregressive coefficient tensor **A**. FaTA disentangles non-stationary temporal dynamics, sparse cross-variate dependencies, and the physical identities of each variate. This decomposition is achieved by extending beyond the standard Tucker method, incorporating specialized designs to improve the representation of temporal patterns and variable interactions. Specifically, FaTA transforms the temporal factor matrix $T$ using sparse circular convolutions with learnable, sparse vectors. Cross-variate matrix $P$ is enforced with permutation invariance via randomized orthogonal projections, while the inner-variate matrix $Q$ is initialized with a channel-wise average, turning a generic baseline into an interpretable anchor. Furthermore, we incorporate a "Plug-In" option for the temporal factor $T$, allowing us to replace FaTA with existing methods and demonstrate how DeFa improves the performance of current models by uncovering additional temporal patterns hidden in the amplifier dynamics.

Overall, the key contributions of this paper are summarised as follows:

- We revisit the decomposition with **NAILong** strategy on non-stationary time series, providing full-time adaptive temporal interactions.
- We propose **FaTA** module for the forecasting process, which disentangles non-stationary dynamics and high-dimensional interactions in multivariate series.
- We demonstrate **DeFa**'s state-of-the-art performance across benchmarks, also show its long temporal interaction could improve existing methods' performance by a Plug-In strategy.

## 2 RELATED WORK

**Forecasting Methods** LogTrans Li et al. (2019) applies log-sparse self-attention, Informer Zhou et al. (2021) uses probabilistic sparse attention, Autoformer Wu et al. (2021) and FEDformer Zhou et al. (2022) incorporate series decomposition or Fourier modules. Crossformer Zhang & Yan (2023) embeds time series in 2D to capture temporal and inter-variable dependencies. Recent architectures inject the additional formulation. PatchTST Nie et al. (2023) processes channels with

patched tokens. PETformer Lin et al. (2023) inserts masked tokens into the input as the attention on future values. However, DLinear Zeng et al. (2023) outperforms baselines as a decomposed linear forecaster. TiDE Das et al. (2023) uses an all-MLP encoder–decoder that preserves simplicity. SparseTSF Lin et al. (2024b) lies a sparse forecasting with 1k Parameters. To restore their advantage, SAMformer Ilbert et al. (2024a) uses channel-wise Transformer trained with sharpness-aware minimisation. CycleNet Lin et al. (2024a) bridges linear and non-linear paradigms by integrating multi-scale cyclical decomposition. Likewise, ElasTST Zhang et al. (2024) and TimeMixer++ Wang et al. (2024c) produce forecasts invariant to changes in the forecast horizon to integrate temporal decomposition.

## 2.1 DISENTANGLE REPRESENTATIONS

Limited research has emphasised disentangled representation learning for time-series forecasting. CoST Woo et al. (2022) employs contrastive learning to disentangle seasonal and trend factors through separate time-domain and frequency-domain encoders. DiPE-Linear Zhao et al. (2024) introduces a parameter-efficient linear network for long-term forecasting. Similarly, TimeDRL Chang et al. (2023) proposes a self-supervised framework that learns dual-level disentangled embeddings at both timestamp and instance levels via predictive and contrastive objectives.

## 3 PROPOSED DECOMPOSE AND FORECAST METHOD

For multivariate time series forecasting, given a historical observation matrix $X = [\mathbf{x}_1, \ldots, \mathbf{x}_n, \ldots, \mathbf{x}_N]^\top \in \mathbb{R}^{N \times H}$ with $N > 1$ variate vectors in length $H$, **DeFa** aims to predict future time series $Y = [\mathbf{y}_1, \ldots, \mathbf{y}_n, \ldots, \mathbf{y}_N] \in \mathbb{R}^{N \times F}$. in length $F$. In the experiment setup, $H = 96$ is the historical length (or the historical window) and the $F = \{96, 192, 336, 720\}$ is the forecasting length set. That is, we forecast future series via DeFa as $\hat{Y} = \text{DeFa}(X)$, then evaluate the results with the ground truth $Y$. The proposed DeFa framework treats (1) NAILong decomposition and (2) forecasting with FaTA as two separate modules. The decomposition part follows the theoretical validation provided in Appendix C. It shows that the decomposed components could be re-formed and approximated to the historical time series as $\hat{X} \in \mathbb{R}^{N \times H}$:

$$[Amp, NS, R]^\top = \text{Decomp}(X), \\ \hat{X} = Amp \cdot NS + R, \tag{1}$$

where $\cdot$ is the element-wise product with matrixes $Amp, NS, R \in \mathbb{R}^{N \times H}$ is defined as temporal-wise Amplifier, a stationary normal-

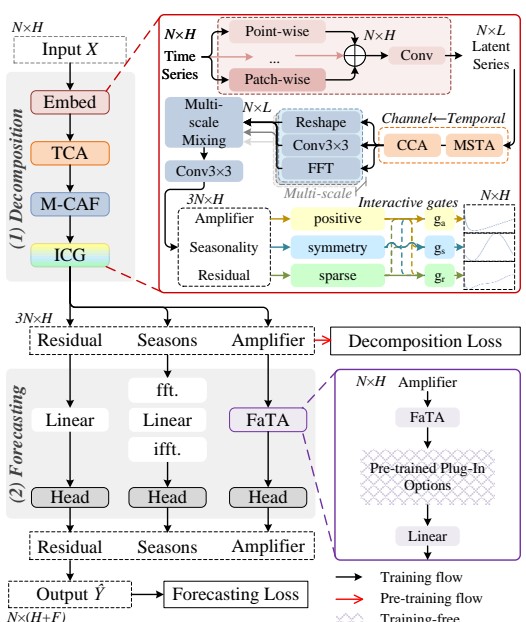

Figure 1: Pipeline of DeFa. DeFa treats (1) decomposition guided by NAILong and (2) forecasting with special FaTA as two modules. Two modules are in the gray in the figure.

ized seasonality, and a sparse residual to random events. Next, by applying individual forecasting layers on decomposed completions, DeFa reforms the future time series $\hat{Y}$ by forecasting individually forecasting components $\hat{Amp} = \text{FaTA}(Amp)$, $\hat{NS}$ and $\hat{R}$ in a special forecast length $(H + F)$:

$$\hat{Y} = \hat{Amp} \cdot \hat{NS} + \hat{R}. \tag{2}$$

The above decomposition is assumed to be built on the univariate time series setting in NAILong theory. Yet this may remain extra challenges on complicated representations for time series forecasting

in each channel when DeFa is assigned with a general multivariate time series input $X \in \mathbb{R}^{N \times H}$. Thus, different to NAILong, FaTA is actually expected to represent various patterns and decompose components for any real-world multivariate time series. Extra details of NAILong and FaTA are provided in Appendix C. **To help the reader to reproduce this work, the overall method descriptions with data flow are introduced in Appendix B**.

### 3.1 MULTI-SCALE NAILONG DECOMPOSITION

**Point- and Patch-wise Embedding & Temporal-Channel Attention (TCA).**   We introduce a compact embedding module designed to enhance temporal representations in the input $X$. It integrates types of embeddings: (1) a positional embedding using a sinusoidal pattern on points; (2) a global context embedding via a linear projection over the full sequence, shared across positions; and (3) a content-aware embedding obtained through a linear transformation followed by repetition along the temporal dimension. This combination efficiently encodes both structural and semantic information with minimal overhead. To enhance both local and global representation within sequential features, we design a two-phase attention module, consisting of Multi-Scale Temporal Attention (MSTA) and Cross-Channel Attention (CCA). This module operates on 1D embedded sequences $\mathbf{X}$. By permuting the layout between $1 \times L$ and $L \times 1$, the model adapts between holistic time-series context and per-point refinement.

**Multi-Component Adaptive Filter (M-CAF).**   Decomposition requires scale-specific processing to handle components: Amplifier manifests at coarse resolutions (long time analysis), Seasonality is best isolated spectrally, and Residuals demand fine-grained analysis (per time stamp). Thus, M-CAF introduce multi-resolution reshaping that creates a resolution pyramid similar to the 8x times downsampling. The inputs include the different scale-reshaped identity from the patch-wise to point-wise, scale-specific local features, and dominant frequencies to isolate pure seasonality. We use concatenation to prevent the components from separating from the superposition. These operations capture the amplified, seasonal, and residual components of the signal. The multi-scale representations are then integrated hierarchically: the feature from a lower level is upsampled and fused with the next higher level using cross-attention and normalization. This process continues recursively until the top level is reached. The final output $\mathbf{x}_{0(\mathrm{up})} \in \mathbb{R}^{B \times 3C \times L}$ is passed through a linear projection layer, producing a tensor $\mathbb{R}^{B \times 3C \times H}$ in the historical window length $H$. Each $C$-channel slice represents decomposed Amplifier ($Amp$), normalized Seasonality ($NS$), or sparse Residuals ($R$).

**Interactive Component Gate (ICG).**   To model domain-specific component interactions, ICG enables an interpretable and physically grounded decomposition of time series signals into distinct dynamic structures. An adaptive gated fusion module dynamically modulates the contribution of each component. The component features are concatenated along the channel dimension and then are performed by a gated cross-fusion to capture interdependencies, via learnable linear projections. Each decomposed temporal component corresponds to a specialized representation, as motivated in our theoretical validation. The *amplitude expert* enforces non-negativity and boundedness through a sigmoid activation (where **Amp** $> 0$). The *seasonality expert* captures zero-mean oscillatory patterns via a bounded $\tanh$ activation and a symmetry enforcement to make value $\mathbf{s}(n) \in [-1, 1]$ at any time $n > 0$. The *residual expert* models abrupt fluctuations and stochastic noise using a soft-threshold operation $\lambda = 0.1$ to achieve a sparse residual mask on small values. Here the amplifier is constrained to low-frequency dynamics via exponential smoothing, ensuring monotonic trends align with domain-specific rates. Seasonality undergoes partial normalization to stabilize stationary patterns while retaining multiplicative coupling to the amplifier. Residuals are sparsified, filtering predictable short-term fluctuations into the amplifier and leaving only true anomalies of random time events.

### 3.2 TENSOR FACTORIZATION ON AUTOREGRESSIVE COEFFICIENT.

For the normalized seasonality $NS$ and the sparse residual $R$, a linear layer is applied for each forecasting process. Considering the potential frequency domain drift remaining in the seasonality, the normalized seasonality is transformed into the frequency domain by Fast Fourier Transform. The spectrum and the amplitude is predicted by one linear layer for each. And one linear layer

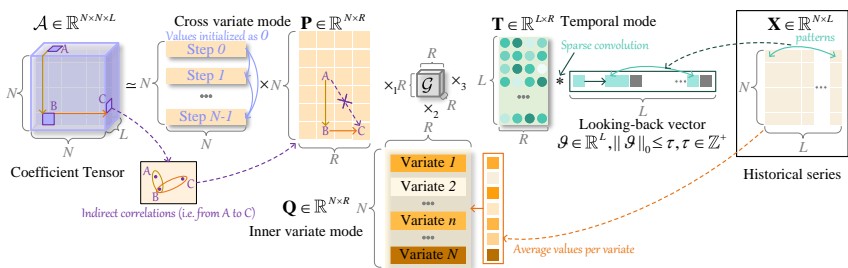

Figure 2: Tensor factorization of the autoregressive coefficient tensor **A** for the Amplifier forecasting. We employ a Tucker decomposition $\mathbf{A} = \mathcal{G} \times_1 P \times_2 Q \times_3 T$ with a core tensor $\mathcal{G}$ and three factor matrices: cross-variate mode $P$, inner-variate mode $Q$, and temporal mode $T$. This design captures both physical interpretability and sparse dynamic patterns across variables and time.

is applied on the sparse residual acts like a mask. Holding adaptive dynamics, $Amp$ encapsulates the most intricate temporal dynamics, including global trends and latent contextual cues to reduce over-smooth (Appendix. C.2). We thus dedicate a specialized forecasting pipeline to the coefficient instead of the series. We re-represent the forecasting towards $\hat{Amp}$ as $\text{FaTA}(Amp) = \mathbf{A} \times_3 Amp$ where $\times_3$ is the third-mode product. To enforce interpretability and parsimony, we factorize $\mathbf{A} \in \mathbb{R}^{N \times N \times (H+F)}$ via Tucker decomposition:

$$\mathbf{A} = \mathcal{G} \times_1 P \times_2 Q \times_3 T,$$
$$\text{s.t.} \quad P^\top P = Q^\top Q = T^\top T = I_R, \tag{3}$$

where the latent factors $\mathbf{P}, \mathbf{Q} \in \mathbb{R}^{N \times R}$ represent low-rank $R$ cross- and inner-variate modes respectively, $\mathbf{T} \in \mathbb{R}^{L \times R}$ represents the temporal mode, and $\mathcal{G} \in \mathbb{R}^{R \times R \times R}$ is the core tensor. The low-rank form $R \ll N$ reduces parameters from $O(N^2 L)$ to $O(R^3 + R(N + N + L))$ where $L = H + F$, facilitating efficient learning. Also, $R$ represents a general understanding of multivariate time series that there are $R$ types of features, i.e. $R = 4$ indicates the periods, seasonality, averages and trends in time series.

**Temporal Mode $T$.** A sparse convolution could help one to understand the time series trends and interpret temporal patterns with located historical time. The sparse regression on the temporal mode provides potential long-term patterns captured from real-world time series. To capture these patterns within one variate, we apply a sparse circular convolution to each time-varying latent vector $\mathbf{t}_r = [t_{1,r}, \ldots, t_{L,r}]^\top, r \in [R]$ among rank $R$. $\mathbf{t}_r = f(\boldsymbol{\vartheta})\mathbf{t}_r$ where $\boldsymbol{\vartheta} = [\theta_1, \ldots, \theta_L]^\top \in \mathbb{R}^L$, s.t. $||\boldsymbol{\vartheta}||_0 \leq \tau, \tau \in \mathbb{Z}^+$ is a learnable sparse filter with at most $\tau$ non-zeros. To achieve the whole temporal mode instead of replying the rank-dimension, this convolution can be equivalently represented as a linear combination $f(\boldsymbol{\vartheta}, F)$ of its sparse parts with an identical cyclic-shift matrix $F$. To model the time series holding shifted or emerging seasonal patterns, we extend the sparse convolution to characterize the time-varying series, or non-stationary series. To improve scalability, we introduce a prediction on the sparse vector $\boldsymbol{\vartheta}$. From the temporal mode $\mathbf{t}_r$ to $\mathbf{t}_{r+1}$, $\boldsymbol{\vartheta}$ is expected to be updated based on the lag. Instead of fixing $\boldsymbol{\vartheta}$ across all latent ranks or time steps, we adapt it dynamically based on the temporal evolution of the time-varying latent matrices $\{\mathbf{t}_r\}_{r=1}^R$. Specifically, for each rank operation $f(\boldsymbol{\vartheta}_r)\mathbf{t}_r$, we predict the temporal embedding $f(\boldsymbol{\vartheta}_{r+1})\mathbf{t}_r$ at rank $r+1$ by updating the sparse filter $\{\boldsymbol{\vartheta}\}_{r \in [R]}$ conditioned on the historical filter.

**Multivariate Modes $Q$ and $P$.** $Q$ encodes variate-specific physical properties and operates as an independent physical identity that enforces domain-dependent physical constraints unique to each variate and makes the temporal mode obey physiological bounds. In this case, the subvector $\mathbf{q}_n$ could be initialized based on the average value of each variate $\text{avg}(\mathbf{amp}_n)$. Variable indices are arbitrary in multivariate series, which indicates $P$ is a permutation-invariant matrix. A new listing order of variates in data would not affect their real-world correlations. To ensure permutation invariance, we project inputs $Amp$ using a random orthogonal matrix $E \in \mathbb{R}^{N \times N}$ as a transformation that permutes variables without altering pairwise distances. We then apply autoregression on $\hat{Amp}$ and invert the projection in the cross-variate mode $P$. During the training, we find that the time-invariant further validates a possible variate-activation forecasting back on current observations.

Table 1: **Seven standard datasets** in the Time Series Pile taken from. ETT Train/validation/test splits in 60%/20%/20% and others in 70%/10%/20% ratio.

| Dataset | Domain | # Channels | Series Length | Dataset Size | Frequency | Forecast Horizon |
|---------|--------|-----------|---------------|--------------|-----------|------------------|
| ETTm1 | Power | 7 | 69680 | (34465, 11521, 11521) | 15 Minute | {96, 192, 336, 720} |
| ETTm2 | Power | 7 | 69680 | (34465, 11521, 11521) | 15 Minute | {96, 192, 336, 720} |
| ETTh1 | Power | 7 | 17420 | (8545, 2881, 2881) | Hourly | {96, 192, 336, 720} |
| ETTh2 | Power | 7 | 17420 | (8545, 2881, 2881) | Hourly | {96, 192, 336, 720} |
| Weather | Weather | 21 | 52696 | (36792, 5271, 10540) | 10 Minute | {96, 192, 336, 720} |
| Electricity | Power | 321 | 26304 | (18317, 2633, 5261) | Hourly | {96, 192, 336, 720} |
| Traffic | Traffic | 862 | 17544 | (12185, 1757, 3509) | Hourly | {96, 192, 336, 720} |

## 3.3 FORECASTING LOSS IN TRAINING

We recall that DeFa plays a decomposition and forecasting pipeline for time series forecasting tasks, and the final time series $\hat{Y}$ are reformulated from predicted components. During the training, MSE loss is applied where DeFa has historical sub-series $\hat{Y}(1:H)$ acting as a constraint with the original observation $X$, ensuring the decomposition modules retain fidelity to observed dynamics without over-fitting to future time:

$$\mathcal{L} = \frac{1}{H+F}\big(\underbrace{||X - \hat{Y}(1:H)||_2^2}_{\text{preserved}} + \underbrace{||Y - \hat{Y}(H+1:H+F)||_2^2}_{\text{forecasting}}\big). \tag{4}$$

## 4 EXPERIMENTS

### 4.1 EXPERIMENTAL SETUP

**Datasets.** To comprehensively assess our DeFa effectiveness over extended periods, we performed experiments on 7 widely-used real-world datasets. (1-4) **ETT** includes four subsets related to electricity transformers. ETTm1 and ETTm2 record the values every 15 minutes. ETTh1 and ETTh2 have hourly recordings. (5) **Weather** consists of 21 climate features at each time step for over thousands of climate recording locations. (6) **Electricity** contains hourly electricity consumption of 321 customers. (7) **Traffic** is an hourly dataset from California transportation department, and consists of road occupancy rates measured on freeways. **Data Splits.** ETT Train/validation/test split follows a 60%/20%/20% segmentation ratio and other dataset follows a 70%/10%/20% ratio. We set the length of the history window to 96 and the forecasting length varies among 96,192,336,720.

**Baseline Methods.** We compare against state-of-the-art models in recent years. (1) **Linear**: NLinear Zeng et al. (2023), DLinear Zeng et al. (2023) and RLinear Li et al. (2023a). (2) **Transformer**: iTransformer Liu et al. (2024b), TimeXer Wang et al. (2024d) and TimeMixer++ Wang et al. (2024a). (3) **Frequency and Others**: CycleNet Lin et al. (2024a), xPatch Stitsyuk & Choi (2025), FreDF Wang et al. (2025), TimeFilter Hu et al. (2025), TQNet Lin et al. (2025), and CMoS Si et al. (2025).

**Evaluation Metrics.** Different with the training loss formulation, the results are demonstrated in terms of mean squared error (MSE) and mean absolute error (MAE). They are calculated from the predicted value $\hat{X}$ and the given future time series $X$ each with $F$ elements. $\text{MSE} = \frac{1}{F}\sum_{i=1}^{F}(\mathbf{x}_i - \hat{\mathbf{x}}_i)^2$ and $\text{MAE} = \frac{1}{F}\sum_{i=1}^{F}|\mathbf{x}_i - \hat{\mathbf{x}}_i|$, where $\mathbf{x}_i$ and $\hat{\mathbf{x}}_i$ are the ground truth value and predicted value at $i$th index of the time.

**Implementation Details.** All experiments in this paper are run five times with random seeds 12,13,42,43,2025, implemented in Python and PyTorch. The training is set with a learning rate of 2e-3, and optimised by the Adam optimiser on NVIDIA GeForce RTX 4090 GPUs (24GB). The batch size is set to be 32. For baselines under the same experimental settings, we follow their implementation details and settings provided in official repositories and original papers. FaTA is set with $\tau = 4$, where MSE score on the temporal mode is less than 5e-4.

Table 2: **Long-term forecasting results** over horizons (96, 192, 336, 720). They are averaged from runs with random seeds (12, 13, 42, 43, 2025). **Bold**: best, Underline: 2nd best. The standard deviation for results is ±0.001. $\tau = 4$.

| Model | ETTh1 | | ETTh2 | | ETTm1 | | ETTm2 | | Weather | | Electricity | | Traffic | |
|---|---|---|---|---|---|---|---|---|---|---|---|---|---|---|
| | MSE | MAE | MSE | MAE | MSE | MAE | MSE | MAE | MSE | MAE | MSE | MAE | MSE | MAE |
| DeFa | **0.404** | **0.417** | 0.337 | 0.369 | **0.325** | **0.365** | **0.250** | 0.305 | 0.230 | 0.264 | **0.164** | 0.255 | **0.408** | 0.275 |
| NLinear | 0.413 | 0.425 | 0.348 | 0.389 | 0.366 | 0.383 | 0.258 | 0.315 | 0.254 | 0.288 | 0.169 | 0.262 | 0.433 | 0.290 |
| DLinear | 0.423 | 0.437 | 0.432 | 0.462 | 0.357 | 0.379 | 0.277 | 0.342 | 0.246 | 0.300 | 0.166 | 0.263 | 0.434 | 0.295 |
| RLinear | 0.413 | 0.418 | 0.351 | 0.347 | 0.364 | 0.385 | 0.256 | **0.286** | 0.264 | 0.265 | 0.171 | 0.272 | 0.435 | 0.409 |
| iTransformer | 0.454 | 0.447 | 0.383 | 0.407 | 0.407 | 0.410 | 0.288 | 0.332 | 0.258 | 0.278 | 0.178 | 0.270 | 0.428 | 0.282 |
| TimeXer | 0.437 | 0.437 | 0.368 | 0.396 | 0.382 | 0.397 | 0.274 | 0.322 | **0.227** | **0.263** | - | - | 0.466 | 0.287 |
| TimeMixer++ | 0.419 | 0.432 | 0.336 | 0.380 | 0.369 | 0.378 | 0.263 | 0.313 | 0.226 | 0.261 | **0.164** | **0.253** | 0.416 | **0.264** |
| CycleNet | 0.432 | 0.431 | 0.384 | 0.404 | 0.386 | 0.395 | 0.272 | 0.314 | 0.254 | 0.278 | 0.170 | 0.260 | 0.485 | 0.312 |
| xPatch | 0.428 | **0.417** | **0.327** | **0.360** | 0.377 | 0.384 | 0.267 | 0.313 | 0.232 | 0.260 | 0.179 | 0.264 | 0.502 | 0.282 |
| FreDF | 0.437 | 0.435 | 0.371 | 0.396 | 0.392 | 0.398 | 0.277 | 0.319 | 0.254 | 0.277 | 0.169 | 0.259 | 0.421 | 0.279 |
| TimeFilter | 0.409 | 0.428 | 0.369 | 0.400 | 0.377 | 0.399 | 0.273 | 0.323 | 0.241 | 0.270 | 0.168 | 0.258 | 0.411 | 0.278 |
| TQNet | 0.443 | 0.436 | 0.377 | 0.402 | 0.378 | 0.398 | 0.280 | 0.326 | 0.252 | 0.269 | 0.168 | 0.258 | 0.445 | 0.276 |
| CMoS | 0.422 | 0.427 | 0.380 | 0.403 | 0.384 | 0.396 | 0.277 | 0.321 | 0.256 | 0.273 | 0.169 | 0.257 | 0.479 | 0.355 |

Table 3: **Plug-In options** are applied for long-term forecasting results. Settings remain the same with Tab. 2. **Bold**: best, Underline: 2nd best. Corrections highlight superior improvements of longer forecasting length, indicating the full-time interactions provided by the FaTA modes.

| Plug-in Options | ETTh1 | | ETTh2 | | ETTm1 | | ETTm2 | | Weather | | Electricity | | Traffic | |
|---|---|---|---|---|---|---|---|---|---|---|---|---|---|---|
| | MSE | MAE | MSE | MAE | MSE | MAE | MSE | MAE | MSE | MAE | MSE | MAE | MSE | MAE |
| DeFa | 0.404 | 0.417 | 0.337 | 0.369 | 0.325 | 0.365 | 0.250 | 0.305 | 0.230 | 0.263 | 0.164 | 0.255 | 0.408 | 0.275 |
| +NLinear | 0.408 | 0.418 | 0.345 | 0.386 | 0.356 | 0.376 | 0.254 | **0.287** | 0.246 | 0.284 | 0.168 | 0.260 | 0.423 | 0.287 |
| +DLinear | 0.412 | 0.424 | 0.377 | 0.396 | 0.350 | 0.373 | 0.256 | 0.318 | 0.237 | 0.289 | **0.161** | 0.259 | 0.427 | 0.290 |
| +RLinear | 0.410 | 0.420 | 0.345 | 0.364 | 0.351 | 0.384 | 0.284 | 0.320 | 0.248 | 0.276 | 0.169 | 0.282 | 0.428 | 0.309 |
| +iTransformer | 0.401 | 0.419 | 0.354 | 0.358 | 0.366 | 0.369 | 0.267 | 0.311 | 0.243 | 0.258 | **0.161** | 0.254 | **0.400** | 0.269 |
| +TimeXer | 0.408 | 0.418 | 0.336 | 0.377 | 0.335 | 0.369 | 0.271 | 0.312 | 0.224 | 0.251 | - | - | 0.404 | 0.282 |
| +TimeMixer++ | **0.399** | 0.412 | **0.313** | 0.360 | **0.323** | **0.360** | **0.245** | 0.302 | 0.223 | 0.250 | 0.161 | **0.247** | 0.399 | **0.262** |
| +CycleNet | 0.426 | 0.429 | 0.384 | 0.405 | 0.381 | 0.399 | 0.266 | 0.313 | 0.244 | 0.269 | 0.167 | 0.258 | 0.480 | 0.285 |
| +xPatch | 0.406 | **0.398** | 0.336 | **0.342** | 0.358 | 0.365 | 0.279 | 0.311 | 0.244 | 0.266 | 0.188 | 0.270 | 0.477 | 0.296 |
| +FreDF | 0.416 | 0.413 | 0.390 | 0.416 | 0.370 | 0.408 | 0.291 | 0.320 | 0.267 | 0.291 | 0.178 | 0.272 | 0.443 | 0.293 |
| +TimeFilter | 0.414 | 0.426 | 0.359 | 0.384 | 0.347 | 0.378 | 0.265 | 0.315 | 0.237 | 0.270 | 0.167 | 0.259 | 0.416 | 0.284 |
| +TQNet | 0.417 | 0.428 | 0.370 | 0.390 | 0.354 | 0.381 | 0.268 | 0.319 | 0.240 | 0.272 | 0.166 | 0.258 | 0.421 | 0.286 |
| +CMoS | 0.426 | 0.417 | 0.379 | 0.404 | 0.364 | 0.388 | 0.273 | 0.318 | 0.246 | 0.277 | 0.170 | 0.260 | 0.440 | 0.292 |

## 4.2 MAIN RESULTS

**Long-Term Forecasting Results.** As shown in Table 2, DeFa achieves competitive performance across seven widely-used time series forecasting benchmarks. It attains the best average MSE/MAE on four datasets (ETTh1, ETTm1, ETTm2, and Traffic) and ranks second on ETTh2, demonstrating consistent superiority over a range of strong baselines.

When compared to linear variants such as NLinear, DLinear, and RLinear, DeFa exhibits stronger predictive capability across most settings, particularly on ETTh1 and ETTm2. Against modern Transformer-based models including iTransformer, TimeXer, and TimeMixer++, DeFa not only maintains competitive accuracy but does so with significantly improved parameter efficiency. Furthermore, compared to recent specialized architectures like CycleNet, xPatch, and FreDF, DeFa shows robust performance, especially on real-world datasets such as Traffic and Electricity. Finally, when contrasted with other contemporary methods like TimeFilter, TQNet, and CMoS, DeFa continues to demonstrate leading results, underscoring its general effectiveness and stability in long-term time series forecasting.

**Plug-In Options.** As shown in Table 3, we further evaluate DeFa as a plug-in module integrated into a variety of existing forecasting models. The results highlight three major advantages. First, DeFa consistently enhances the temporal modeling capability of base architectures, with particularly notable improvements on long-horizon forecasting tasks. Transformer-based models benefit the most from this integration, often exceeding the performance of both standalone DeFa and linear models under the same decomposition setup, suggesting that DeFa effectively amplifies the ability of these models to capture complex periodic structures. Second, DeFa serves as a lightweight and

Table 4: Ablation Study of **DeFa** forecasting with historical length $W = 96$ on ETT-series dataset.

| $F$ | Original DeFa | | w/ $\mathbb{A}$ | | w/ $\mathbb{B}$ | | w/ $\mathbb{C}$ | | w/ $\mathbb{D}$ | | w/ $\mathbb{E}$ | | w/ $\mathbb{F}$ | |
|---|---|---|---|---|---|---|---|---|---|---|---|---|---|---|
| | MSE | MAE | MSE | MAE | MSE | MAE | MSE | MAE | MSE | MAE | MSE | MAE | MSE | MAE |
| 96 | 0.274 | 0.319 | 0.273 | 0.319 | 0.273 | 0.320 | 0.816 | 1.07 | 1.317 | 1.573 | 0.272 | 0.317 | 0.273 | 0.316 |
| 192 | 0.308 | 0.349 | 0.309 | 0.351 | 0.311 | 0.350 | 1.34 | 1.29 | 1.602 | 1.948 | 0.306 | 0.347 | 0.307 | 0.349 |
| 336 | 0.342 | 0.373 | 0.344 | 0.371 | 0.343 | 0.375 | 1.82 | 1.73 | 2.527 | 2.910 | 0.340 | 0.372 | 0.340 | 0.374 |
| 720 | 0.392 | 0.416 | 0.394 | 0.418 | 0.395 | 0.417 | 2.29 | 2.53 | 3.013 | 4.392 | 0.390 | 0.415 | 0.391 | 0.414 |

efficient preprocessor that decomposes time series into seasonal and residual components before feeding them into subsequent modules. This decomposition requires only minimal computational overhead—for instance, when augmented with DeFa, TimeMixer++ achieves better performance on ETTh1 with only a modest increase in memory usage. This illustrates the practical efficiency of DeFa in real-world applications. Third, the plug-in version of DeFa delivers substantial performance gains across multiple model families, including linear, Transformer, and multi-scale architectures, with only a slight increment in computational cost. This makes it a highly effective and economical strategy for enhancing existing forecasting systems without architectural redesign.

**Efficiency Analysis.** Table 5 shows that DeFa runs at 0.11s/iteration and uses just 0.57GB GPU memory, 4 times faster and 6 times lighter than Transformer baselines. While Linear models are marginally faster, DeFa stays within 2 times cost and significantly improves accuracy. It also uses over 10 times less memory than frequency-aware models like FreDF, avoiding memory bottlenecks. DeFa's structured decomposition and sparse convolution make it both scalable and practical for long-sequence forecasting.

Table 5: **Efficiency analysis** on ETT-series dataset under 96-96 forecasting of FaTA, current approaches, and their Plug-Ins (Batch size = 32, FaTA is set with $\tau = 4$).

| Models (wo/w DeFa) | Training Time (s/iter) | Memory Footprint GB |
|---|---|---|
| DeFa | 0.11 | 1.57 |
| NLinear | 0.06/0.15 | 0.64/2.13 |
| DLinear | 0.05/0.14 | 0.65/2.15 |
| RLinear | 0.06/0.16 | 0.64/2.13 |
| iTransformer | 0.29/0.40 | 2.92/4.60 |
| TimeXer | 0.49/0.59 | 4.71/6.43 |
| TimeMixer++ | 0.17/0.28 | 2.77/4.42 |
| CycleNet | 0.26/0.37 | 0.94/2.69 |
| xPatch | 0.33/0.44 | 0.81/2.54 |
| FreDF | 0.26/1.37 | 3.10/4.74 |
| TimeFilter | 0.13/0.23 | 0.58/2.07 |
| TQNet | 0.33/0.45 | 0.61/2.10 |
| CMoS | 0.26/1.36 | 1.14/2.88 |

### 4.3 ABLATION STUDY

**Decomposition Performance with NAILong.**
*How much do decomposition designs match the theoretical validation?*

To quantify how NAILong aligns with its theoretical design, the comparisons of decomposed results are set up between 'historical+future' series and predicted series. In the training, DeFa predicts series in both historical and future time at a length $W + F$. We evaluate NAILong in (1) historical series preservations ($W$), (2) Amplifier ($W + F$), (3) Seasonality's spectrum in frequency domain ($W + F$) and (4) Residual ($W + F$).

As shown in Tab. 6, the results confirms that **Amp**'s role in preserving interactions despite input length variations, **NS** achieves near-perfect reconstruction and **R** maintains low errors on sparse noise isolation. These results affirm that DeFa's multiplicative decomposition operates as theoretically intended. Critical patterns are isolated upfront even in historical time.

Table 6: Ablation Study of **NAILong** decomposition with historical length $W = 96$ on ETT-series dataset.

| $W + F$ | X and $\hat{Y}(1 : H)$ | | Amp | | NS (FFT) | | R | |
|---|---|---|---|---|---|---|---|---|
| | MSE | MAE | MSE | MAE | MSE | MAE | MSE | MAE |
| 192 | 0.015 | 0.082 | 0.074 | 0.0129 | 0.001 | 0.002 | 0.016 | 0.027 |
| 288 | 0.017 | 0.095 | 0.081 | 0.0129 | 0.001 | 0.002 | 0.019 | 0.029 |
| 432 | 0.021 | 0.109 | 0.103 | 0.0129 | 0.001 | 0.002 | 0.021 | 0.032 |
| 816 | 0.026 | 0.120 | 0.133 | 0.0129 | 0.002 | 0.002 | 0.029 | 0.035 |

**FaTA Forecasting.** To dissect the contribution of each architectural block, we conduct a systematic ablation study. As shown in Tab. 4, six variants are evaluated against the full DeFa model across forecasting horizons $F$. **Limited pre-training data** ($\mathbb{A}$) and ($\mathbb{B}$): We pre-trained the decomposition module with only first 10% of dataset or with only first 1% of dataset. The results confirm

the decomposition module's ability to learn generalizable temporal patterns with minimal supervision. **Architectural Simplicity** (ℂ): We replace the whole decomposition module by three Linear layers to achieve the decompose, ignoring every multi-channel or temporal modelling in DeFa. The decrescent proves that effective decomposition requires explicit modelling of multi-channel temporal dependencies validated in NAILong. **Additive vs. NAILong Formulation** (𝔻): We replace the NAILong decomposition $\hat{\mathbf{X}}_h = \mathbf{Amp} \cdot \mathbf{NS} + \mathbf{R}$ by a simple addition as $\hat{\mathbf{X}}_h = \mathbf{Amp} + \mathbf{NS} + \mathbf{R}$. The replacement catastrophically degrades performance, empirically validating our theoretical claim that proposed NAILong is better with the adaptive interactions. **Forecaster Design** (𝔼) and (𝔽): we replace the forecasting linear layer by a Transformer decoder in the forecasting process of $\mathbf{NS}$ and $\mathbf{R}$ respectively. Transformer-based forecasting of $\mathbf{NS/R}$ yields marginal gains but increases training time over 10 times in our study, suggesting linear layers optimally balance performance and efficiency for amplitude-conditioned components.

**Impact of Different Historical Length $H$.** As illustrated in Fig. 3, we systematically examine the influence of historical observation length $H$ on forecasting accuracy across varying prediction horizons $F$. In contrast to attention-based models—which often degrade with excessively long historical windows due to distraction from irrelevant signals—DeFa consistently benefits from extended input sequences, achieving monotonic improvements in accuracy up to $F = 720$. Performance only declines when $H < F$, a result of insufficient contextual information for long-horizon extrapolation. However, once $H \geq F$, DeFa's decomposition mechanism robustly captures long-range temporal patterns without overfitting, effectively condensing extended histo-

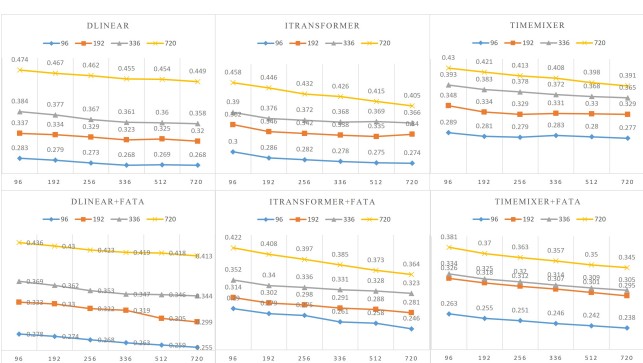

Figure 3: **Ablation study** on different historic series lengths (96, 192, 256, 336, 512, 720) to different forecasting lengths (96, 192, 336, 720). The trends demonstrate that FaTA's modes help other methods improve and extend the performance when using historical windows longer than the forecasting length.

ries into structured and compact representations. These findings challenge the conventional practice of restricting $H$ to match $F$, and underscore DeFa's capability to filter noise through implicit structural decomposition, making it especially suitable for real-world forecasting tasks that require effective long-memory modeling.

## 5 CONCLUSION

In this paper, we propose DeFa, a unified decomposition-and-forecast framework designed to address the challenges of non-stationary and multi-scale dynamics in multivariate time series forecasting. By introducing the NAILong decomposition strategy, DeFa adaptively disentangles time series into interpretable components—a time-varying amplifier, stationary seasonal patterns, and sparse residuals. To further capture complex temporal and cross-variable dependencies, we develop the FaTA module, which applies a factorized tensor autoregression mechanism to efficiently model high-dimensional interactions while improving both accuracy and scalability. Extensive experiments demonstrate that DeFa achieves state-of-the-art performance across a range of real-world benchmarks. Moreover, its plug-in capability shows that the proposed decomposition and forecasting mechanisms can be effectively integrated into existing models, enhancing their ability to handle long-term temporal dependencies and non-stationary patterns.

## 6 REPRODUCIBILITY STATEMENT

Since we proposed novel models and algorithms, we share a link to the anonymous GitHub webpage https://anonymous.4open.science/r/TSF-DB46/README.md with download-

able source code. For theoretical validations of NAILong, we provide clear explanations of assumptions and a complete proof of the claims in the appendix. We also provide a very detailed data flow method description in the appendix for the reader to reproduce our work. For the public datasets we used in the experiments, we provide a complete description of the data in the experiment section, and we provide the code with detailed processing steps.

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

# Support Material
## of
## DeFa

## A  LIMITATION AND FUTURE WORK.

**Limitation**

- DeFa was evaluated as a novel decomposition-forecasting method on time series forecasting tasks. Designed as a framework supporting backbone plug-in, this paper left potential evaluations across other time series downstream tasks, such as classifications, imputations, and zero-shot tasks.

- All experiments and evaluations were limited by some specific time-series dataset. There are a large amount of time series in the form of non-stationary data that we could use to prove the effectiveness of DeFa in the industrial domain.

- DeFa relied on a certain length of historical time series to predict future time series. Designed as a channel-independent forecasting, DeFa's forecasting module may encounter challenges when the number of channels/attributes in time series is too large (i.e. stack trading of 1,000 companies).

- DeFa's performance real-world dataset did not implies its functions on any synthesized time series, i.e. the residual is the Gaussian white noise. The performance may drop when DeFa meet time series hold no real-world meanings.

- Where the foundation formulation of LTSF methods was built on the historical observations, they would perform as less expected on data with massive random events than usual real-word scenarios where $\mathbf{R}$ is not pure white noise but is assumed as a real-world residual. Also, if the time series holding dramatical long-range time steps (stacks trading 10,000 times per seconds), it would be a big difficulty for any models in learning the patterns to forecast.

Please be cautious while using DeFa and other mentioned methods for financial investments. Any forecasters would just provide assistance and suggestions to the final decisions. E.g., the existing out-performance from various methods enables the author to explore and forecast billionaire wealth, until the author realizes that nothing is ever predictable when it comes to the tariffs war.

**Future Work**

- There is a great potential to align the time series decomposition (not only NAILong) with interpretable prompts in LLM4TS.

- It is interesting if we consider the time series into spatial-temporal data (Traffic dataset may hold some implicit geographic information itself). Then the foundational problem of discovering interpretable stationary dynamic could be extended as spatial-temporal forecasting. patterns

## B  DETAILED METHODS

We add the method description with further explanations due to the page limit of the main text. To make both the reading and coding more convenient for the reader, we separate the subsections in the decomposition module and the forecasting module. We will redefine and re-mark the input in each part, which is easier to follow on a single page by marking differences.

### B.1  NAILONG DECOMPOSITION

Our proposed decomposition module in Section 3.1 'Decomposition' is the technique based on NAILong. This module enables multiplicative separation of non-stationary signals at the data-driven level.

In time series long-term forecasting, given multivariate historical input $\mathbf{X} = [\mathbf{x}_1, \ldots, \mathbf{x}_h, \ldots, \mathbf{x}_H] \in \mathbb{R}^{C \times H}$, it aims to predict future time series $\mathbf{Y} = [\mathbf{y}_1, \ldots, \mathbf{y}_f, \ldots, \mathbf{y}_F] \in \mathbb{R}^{C \times F}$. $H$ and $F$ refers to the number of time steps in the history and future. $C > 1$ is the number of the attributes in time series, respectively, the number of the channels. In the experiment setup, $H = 96$ as the historical length (or the historical window) and the $F = \{96, 192, 336, 720\}$ as the forecasting length.

As shown in Fig. 1, proposed DeFa framework treats (1) NAILong decomposition and (2) forecasting as two separate modules. NAILong decomposition follows the theoretical validation provided in Appendix C. It shows that the decomposed components could be re-formed and approximated to the historical time series as $\hat{\mathbf{X}} \in \mathbb{R}^{C \times H}$:

$$[\mathbf{Amp}, \mathbf{NS}, \mathbf{R}]^\top = \text{NAILong}(\mathbf{X})$$
$$\hat{\mathbf{X}} = \mathbf{Amp} \odot \mathbf{NS} + \mathbf{R}, \tag{5}$$

where $\mathbf{Amp}, \mathbf{NS}, \mathbf{R} \in \mathbb{R}^{C \times H}$ is defined as temporal-wise Amplifier, a stationary normalized seasonality, and a sparse residual to random events. By applying individual forecasting layers on decomposed completions, DeFa forecasts the future time series $\hat{\mathbf{Y}}$ by individually forecasting components $\hat{\mathbf{Amp}}$, $\hat{\mathbf{NS}}$ and $\hat{\mathbf{R}}$ in forecast length $(H + F)$:

$$\hat{\mathbf{Y}} = \hat{\mathbf{Amp}} \odot \hat{\mathbf{NS}} + \hat{\mathbf{R}}. \tag{6}$$

For the above decomposition assumptions which build on a univariate time series $\mathbf{Z} \in \mathbb{R}^H$ setting in NAILong theory, the hypothesize that an effective non-stationary decomposition can enable a significantly simpler and more efficient forecasting on the decomposed components. Yet this may remain extra challenges on complicated representations for time series in each channel if DeFa is assigned with a multivariate time series input $\mathbf{X} \in \mathbb{R}^{C \times H}$. Here, NAILong is actually expected to represent various patterns and decompose components for any real-world multivariate time series. Extra details of NAILong and decomposed components are provided in Appendix C.

*The training designs follow ideas in NAILong.*

As NAILong validation starts from time-related formulations, **embeddedding** first add time series with temporal information per patch (from each point to whole length). And we perform the relationship in multivariate data in the latent space by attention mechanisms. The next two modules, **Multi-Component Adaptive Filter** and **Interactive Component Gate**, perform the adaptations in the components and then their interactions, as the validation perform such transforms between components and apply constraints on each component.

### B.1.1 MULTI-SCALE NAILONG

**Embedding.** We introduce a compact embedding module designed to enhance temporal representations in the input $\mathbf{X}_h$. It integrates four types of embeddings via residual addition: (1) a positional embedding using a sinusoidal pattern on points; (2) a global context embedding via a linear projection over the full sequence, shared across positions; and (3) a content-aware embedding obtained through a linear transformation followed by repetition along the temporal dimension. This combination efficiently encodes both structural and semantic information with minimal overhead. A 2D convolution mapped the embedded input to $\mathbf{X}$, from length H to a fixed latent sequence length (typically L = 512).

**Attention.** To enhance both local and global representation within sequential features, we design a two-phase attention module, consisting of Multi-Scale Temporal Attention (MSTA) and Cross-Channel Attention (CCA). This module operates on 1D embedded sequences $\mathbf{X}$. By permuting the layout between $1 \times L$ and $L \times 1$, the model adapts between holistic time-series context and per-point refinement.

**Multi-Scale Temporal Attention (MSTA).** Time series exhibit patterns at multiple granularities (e.g., hourly fluctuations vs. weekly trends). Standard single-scale convolutions fail to capture this hierarchy. To explicitly correlate with physical scales, MSTA first addresses a multi-kernel convolution to extract scale-specific features, from sensor noises as short-term spikes (k=3) to cycles as periods (k=9). To preserve inter-granular interactions, features with different time-scales (hourly, weekly) are fused via a channel-wise concatenation $\oplus$. To suppress transient noise and learn physical

weights from the dataset scenario, a dynamic gating $\odot$TimeAvg is applied using the global average. The residual connection is introduced to ensure information recovery:

$$\mathbf{X}_{\text{MSTA}} = \sigma(\text{LayerNorm}(\text{Conv1d}_{1\times1}(\oplus_k\text{Conv1d}_k(\mathbf{X}))) \odot \text{TimeAvg}(\mathbf{X})) + \mathbf{X}. \quad (7)$$

**Cross-Channel Attention (CCA).** To handle different multivariate sets in different datasets, we introduce CCA to model the cross-variate coupling and refine the temporal features without constraints from the variate dimension. CCA adapts average pooling on two dimensions to capture temporal context ($avg_t$) and cross-variate dependencies ($avg_c$). The residual link would preserve locality in temporal features (dim=2). The 1x1 Convolution is applied to refine the pooled temporal features:

$$\mathbf{X}_{\text{avg-t}} = \text{Conv1d}_{1\times1}(\text{AAP}(\mathbf{X}_{\text{MSTA}}, \text{dim} = 2) + \mathbf{X}_{\text{MSTA}}), \ \mathbf{X}_{\text{avg-c}} = \text{AAP}(\mathbf{X}_{\text{MSTA}}, \text{dim} = 1). \quad (8)$$

The dot $\cdot$ represents the MSTA output. As temporal patterns (MSTA) and cross-variable couplings (CCA) interact multiplicatively in real systems, we integrate the Harmard Product $\odot$ to fuse temporal features and channel-wise averages. The multiplication would enforce conditional modulations where CCA features gate MSTA features, aligning with physical systems. The additive residual ensures local temporal features anchor the fusion, preventing over-smoothing at the peaks. The full output of the attention mechanism integrates both phases through residual summation:

$$\underbrace{(\mathbf{X}_{\text{attn}} + \mathbf{X})}_{\text{MSTA}} \to \underbrace{(\mathbf{X}_{\text{avg-t}} \odot \mathbf{X}_{\text{avg-c}} + \cdot)}_{\text{CCA}},$$
$$\mathbf{X} = \mathbf{X}_{\text{avg-t}} \odot \mathbf{X}_{\text{avg-c}} + \mathbf{X}_{\text{MSTA}}. \quad (9)$$

**Multi-Component Adaptive Filter (M-CAF).** Decomposition requires scale-specific processing to handle components: Amplifier manifests at coarse resolutions (long time analysis), Seasonality is best isolated spectrally, and Residuals demand fine-grained analysis (per time stamp). Thus, M-CAF introduce multi-resolution reshaping that creates a resolution pyramid similar to the 8x times downsampling. The inputs include the different scale-reshaped identity from the patch-wise to point-wise, scale-specific local features, and dominant frequencies to isolate pure seasonality. We use concatenation $\oplus$ to prevent the components from separating from the superposition. To decompose time series into three components, M-CAF processes various aspects of the input sequence, enabling the model to adaptively extract multi-scale temporal features. The input tensor $\mathbf{X} \in \mathbb{R}^{(B\times C)\times 1\times L}$ is reshaped into $\mathbf{X}_l \in \mathbb{R}^{B\times 8^l C\times 8^{-l}L}$ to simulate clip-wise learning at $l$th level $l = 1, \ldots, L$. On this reshaped tensor, we apply 1D convolutions with different kernel sizes $k_j$, followed by concatenation and a Fourier transform along the temporal axis:

$$\mathbf{X}_{l(\text{down})} = \mathbf{X}_l \oplus \text{Conv1d}_{1\times1}(\oplus_{j=1}^6\text{Conv1d}_{k_j}(\mathbf{X}_l)) \oplus \text{FFT}(\mathbf{X}_l), \ \mathbf{X}_{l(\text{down})} \in \mathbb{R}^{B\times(3\cdot8^l C)\times 8^{-l}L}. \quad (10)$$

These operations capture the amplified, seasonal, and residual components of the signal. The multi-scale representations are then integrated hierarchically: the feature from a lower level is upsampled and fused with the next higher level using cross-attention and normalization:

$$\mathbf{X}_{l-1(\text{up})} = \text{LayerNorm}(\text{Cross-Attn}(\text{Conv1d}_{Up}(\mathbf{X}_{l(\text{up})}), \mathbf{X}_{l-1(\text{down})}, \mathbf{X}_{l-1(\text{down})})). \quad (11)$$

This process continues recursively until the top level is reached. The final output $\mathbf{x}_{0(\text{up})} \in \mathbb{R}^{B\times3C\times L}$ is passed through a linear projection layer, producing a tensor $\mathbb{R}^{B\times3C\times H}$ in the historical window length $H$. Each $C$-channel slice represents decomposed Amplifier (**Amp**), normalized Seasonality (**NS**), or sparse Residuals (**R**).

**Interactive Component Gate (ICG).** To model domain-specific component interactions, ICG employs a SoftMax-gated specialisation of M-CAF outputs to provide weights as the sustained trend $g_s$ on Amplifier **Amp**, seasonal peaks $g_s$ on Seasonality **S**, and anomaly event-driven disruptions $g_r$ on Residual **R**. Then, ICG dynamically modulates cross-component interactions, where the interactions between components would be adaptive due to different time scales. Assumed in NAILong, the Seasonality and Amplifier hold multiplicative couplings observed in real data. We capture interactions via learnable linear projections $f_\theta, f_\phi$ on **Amp** and **S** respectively. **R** would

anchor the unexplained information to prevent over-sparsify restore and to absorb the anomality in $\mathbf{Amp}$ and $\mathbf{S}$:

$$\mathbf{G} = \mathrm{softmax}(\mathbf{Amp} \oplus \mathbf{NS} \oplus \mathbf{R}), \quad [g_a, g_s, gr] = \mathbf{G} \in \mathbb{R}^{B \times 3 \times C},$$
$$\mathbf{Amp} = g_a \mathbf{Amp} + g_s f_\theta(\mathbf{NS}, \mathbf{R}), \ \ \mathbf{NS} = g_s \mathbf{NS} + g_a f_\phi(\mathbf{Amp}, \mathbf{R}), \tag{12}$$
$$\mathbf{R} = g_r \mathbf{R} + \frac{1}{2}(g_a \mathbf{Amp} + g_s \mathbf{NS}).$$

Each decomposed temporal component corresponds to a specialized representation, as motivated in our theoretical validation. The *amplitude expert* enforces non-negativity and boundedness through a sigmoid activation (where $\mathbf{Amp} > 0$). The *seasonality expert* captures zero-mean oscillatory patterns via a bounded $\tanh$ activation and a symmetry enforcement to make value $\mathbf{s}(n) \in [-1, 1]$ at any time $n > 0$. The *residual expert* models abrupt fluctuations and stochastic noise using a soft-threshold operation $\lambda = 0.1$ to achieve a sparse residual mask on small values:

$$\mathbf{Amp} = \sigma(\mathbf{Amp}), \quad \mathbf{NS} = \tanh(\mathbf{NS}) - \frac{1}{T}\sum_{t=1}^{T} \tanh(\mathbf{NS}), \quad \mathbf{R} = \mathrm{sign}(\mathbf{R}) \cdot \max(|\mathbf{R}| - \lambda, 0), \tag{13}$$

here the amplifier is constrained to low-frequency dynamics via exponential smoothing, ensuring monotonic trends align with domain-specific rates. Seasonality undergoes partial normalization to stabilize stationary patterns while retaining multiplicative coupling to the amplifier. Residuals are sparsified, filtering predictable short-term fluctuations into the amplifier and leaving only true anomalies of random time events.

### B.1.2 DECOMPOSITION LOSS IN PRE-TRAINING

In the pre-training stage, all training data are used to optimize the decomposition module within 3 epochs. We discover that the pre-training could improve the forecasting performance a bit and much lower the training cost on GPU memory and the training time. Ideally in real-valued scenarios, NAILong could exactly decompose the underlying trend, seasonal structure, and stochastic noise components. Thus we define a decomposition loss for the pre-training stage or the training stage:

$$\mathcal{L}_d = \frac{1}{H}||\mathbf{X} - \mathbf{Amp} \odot \mathbf{NS} \oplus \mathbf{R}||_2^2. \tag{14}$$

Here, an interpretable mode is connected with the network training, where real-world data expressions are added to the decomposed components.

### B.2 FORECATING WITH FATA

We consider forecasting $N$ interdependent variables over $L$ time steps. Given historical observations $\mathbf{X} \in \mathbb{R}^{N \times L}$, our goal is to predict future values $\mathbf{Y} \in \mathbb{R}^{N \times L}$. We model this task as a tensorized autoregression:

$$\mathbf{Y} = \mathcal{A} \times_3 \mathbf{X} + \boldsymbol{\epsilon}, \tag{15}$$

where $\mathcal{A} = \{\mathbf{A}_t\}_{t \in [L]} \in \mathbb{R}^{N \times N \times L}$ is a third-order tensor whose slice $\mathbf{A}_l$ encodes variable interactions at lag $l$. This formulation allows us to capture both multivariate dependencies and time-varying patterns in a unified framework.

### B.3 TENSOR FACTORIZATION ON AUTOREGRESSIVE COEFFICIENT

To enforce interpretability and parsimony, we factorize $\mathcal{A}$ via Tucker decomposition:

$$\mathcal{A} = \mathcal{G} \times_1 \mathbf{P} \times_2 \mathbf{Q} \times_3 \mathbf{T},$$
$$\text{s.t.} \quad \mathbf{P}^\top \mathbf{P} = \mathbf{Q}^\top \mathbf{Q} = \mathbf{T}^\top \mathbf{T} = \mathbf{I}_R, \tag{16}$$

where the latent factors $\mathbf{P}, \mathbf{Q} \in \mathbb{R}^{N \times R}$ represent cross- and inner-variate modes respectively, $\mathbf{T} \in \mathbb{R}^{L \times R}$ represents the temporal mode, and $\mathcal{G} \in \mathbb{R}^{R \times R \times R}$ is the core tensor. The low-rank form $R \ll N$ reduces parameters from $O(N^2 L)$ to $O(R^3 + R(N + N + L))$, facilitating efficient

learning. Also, $R$ represents a general understanding of multivariate time series that there are $R$ types of features, i.e. $R = 4$ indicates the periods, seasonality, averages and trends in time series. And slice $\mathbf{A}_l$ becomes:

$$\mathbf{A}_l = \mathcal{G} \times_1 \mathbf{P} \times_2 \mathbf{Q} \times_3 \mathbf{t}_l^\top = \mathbf{P}\mathbf{G}(\mathbf{t}_l \otimes \mathbf{Q}^\top), \tag{17}$$

where $\mathcal{G}(1) \in \mathbb{R}^{R \times R^2}$ is the mode-1 unfolding and $\mathbf{t}_l$ is the $l$-th row of $\mathbf{T}$.

## B.4 TEMPORAL MODE $\mathbf{T}$

### B.4.1 SPARSE CONVOLUTION FOR TEMPORAL PATTERNS

Real-world time series often exhibit seasonal and periodic effects. The convolution helps one to understand the time series trends and interpret temporal patterns with located historical time. The sparse regression on the temporal mode provides potential long-term patterns captured from real-world time series. To capture these patterns within one variate, we apply a sparse circular convolution to each time-varying latent matrix $\mathbf{t}_r = [t_{1,r}, \ldots, t_{L,r}]^\top, \quad r \in [R]$ among rank $R$:

$$\mathbf{t}_r \approx \begin{bmatrix} \theta_1 t_{1,r} \\ \cdots \\ \sum_{i \in [L]} \theta_{L-i+1} t_{i,r} \end{bmatrix} = \begin{bmatrix} t_{1,r} \\ \cdots \\ t_{L,r} \end{bmatrix} \star \boldsymbol{\vartheta} = \boldsymbol{\vartheta} \star \mathbf{t}_r = f(\boldsymbol{\vartheta})\mathbf{t}_r, \tag{18}$$

where $\star$ denotes circular convolution. $\boldsymbol{\vartheta} = [\theta_1, \ldots, \theta_L]^\top \in \mathbb{R}^L, \quad \text{s.t.} \quad ||\vartheta||_0 \leq \tau, \tau \in \mathbb{Z}^+$ is a learnable sparse filter with at most $\tau$ non-zeros. To achieve the whole temporal mode instead of re-plying the rank-dimension, this convolution can be equivalently represented as a linear combination of its sparse parts as

$$f(\boldsymbol{\vartheta}) = \theta_1 \mathbf{I} + \theta_2 \mathbf{F} + \theta_3 \mathbf{F}^2 + \cdots + \theta_L \mathbf{F}^{L-1}, \tag{19}$$

where $\mathbf{F}$ is the cyclic-shift operator

$$\mathbf{F} = \begin{bmatrix} 0 & 0 & \cdots & 0 & 1 \\ 1 & 0 & \cdots & 0 & 0 \\ 0 & 1 & \cdots & 0 & 0 \\ \vdots & \vdots & \ddots & \vdots & \vdots \\ 0 & 0 & \cdots & 1 & 0 \end{bmatrix}. \tag{20}$$

### B.4.2 SPARSE CONSTRAINTS ACROSS RANKS

To model the time series holding shifted or emerging seasonal patterns, we extend the sparse convolution to characterize the time-varying series, or non-stationary series. To improve scalability, we introduce a prediction on the sparse vector $\boldsymbol{\vartheta}$. From the temporal mode $\mathbf{t}_r$ to $\mathbf{t}_{r+1}$, $\boldsymbol{\vartheta}$ is expected to be updated based on the lag. Instead of fixing $\boldsymbol{\vartheta}$ across all latent ranks or time steps, we adapt it dynamically based on the temporal evolution of the time-varying latent matrices $\{\mathbf{t}_r\}_{r=1}^R$. Specifically, for each rank operation $f(\boldsymbol{\vartheta}_r)\mathbf{t}_r$, we predict the temporal embedding $f(\boldsymbol{\vartheta}_{r+1})\mathbf{t}_r$ at rank $r+1$ by updating the sparse filter $\{\boldsymbol{\vartheta}\}_{r \in [R]}$ conditioned on the historical filter.

Now we re-denote the rank-fixed sparse vector $\boldsymbol{\vartheta}$ into $\boldsymbol{\vartheta}$ This update mechanism is formulated as:

$$\boldsymbol{\vartheta}^{(r+1)} = \Omega(\mathbf{t}_r, \boldsymbol{\vartheta}^{(r)}), \quad ||\boldsymbol{\vartheta}^{(r+1)}||_0 \leq \tau, \tag{21}$$

where $\Omega(\cdot)$ denotes a learnable update function to preserve the interpretability and sparsity of the convolutional kernel at each step.

**Plug-In Options for Enhanced Temporal Pattern Modelling** The sparse convolutional component of is modular and extensible. We allow plug-in alternatives for $\Omega$ to improve modelling flexibility. For example, a frequency-domain transformer can be integrated to infer dominant periodicities and construct $\{\boldsymbol{\vartheta}\}_{r \in [R]}$ directly via spectral attention.

## B.5 MULTIVARIATE MODES $\mathbf{P}$ AND $\mathbf{Q}$

### B.5.1 MODELLING OF INNER-VARIATE RELATIONS

$\mathbf{Q}$ encodes variate-specific physical properties. In $\mathbf{T} \otimes \mathbf{Q}$, $\mathbf{Q}$ operates as an independent physical identity that enforces domain-dependent physical constraints unique to each variate and makes the temporal mode obey physiological bounds. In this case, the $\mathbf{q}_n$ could be initialized based on the average value of each variate $\mathrm{avg}(\mathbf{x}_n)$.

### B.5.2 PERMUTATION-INVARIANT MULTIVARIATE ORDERING ON $\mathbf{P}$

Variable indices are arbitrary in multivariate series. A new listing order of variates in data would not affect their real-world correlations. To ensure permutation invariance, we project inputs $\mathbf{X}$ using a random orthogonal matrix $\mathbf{E} \in \mathbb{R}^{N \times N}$:

$$\hat{\mathbf{X}} = \mathbf{E}\mathbf{X}, \quad \text{s.t.} \quad \mathbf{E}^\top \mathbf{E} = \mathbf{I}_N, \tag{22}$$

as a transformation that permutes variables without altering pairwise distances. We then apply autoregression on $\hat{\mathbf{X}}$ and invert the projection in the cross-variate mode $\mathbf{P}$:

$$\mathbf{x}_l \simeq (\mathbf{E}^\top \mathbf{P})\mathbf{G}\Big(\big(f(\boldsymbol{\vartheta})\mathbf{t}_l\big) \otimes \mathbf{Q}\Big)\hat{\mathbf{x}}_l. \tag{23}$$

The time-invariant indicates a variate activation on current time:

$$\mathbf{y}_l - \mathbf{x}_l \simeq (\mathbf{E}^\top\big[0, \mathbf{P}[1:]\big])\mathbf{G}\Big(\big(f(\boldsymbol{\vartheta})\mathbf{t}_l\big) \otimes \mathbf{Q}\Big)\hat{\mathbf{x}}_l, \tag{24}$$

where indicates the initialization on $\mathbf{P}$ as 0. And for each next-variate generation, we have

$$y_{l,n+1} = \sigma\Big(\frac{1}{\sqrt{n}} \sum_{i=1}^{n} (\mathbf{p}_i \cdot \frac{1}{\sqrt{L}} \sum_{j=1}^{L} \mathbf{y}_{l,n})\Big), \tag{25}$$

where $\sigma$ indicates the tanh activation function (or could be others with a bit less performance).

## B.6 CORE TENSOR ESTIMATION

The core tensor $\mathcal{G}$ could be interpreted as the unfolding $\{\mathbf{G}_r\}_{r \in [R]}$ as capturing global weights across $R$ presentations, yet without any constraint. The relationships are simplified as a linear transform $\Phi \in \mathbb{R}^{R \times R^2}$:

$$\Phi \mathbf{s}_l^\top (\mathbf{E}^\top \mathbf{G})\Big(\big(f(\mathbf{t}_l)\boldsymbol{\vartheta}^\top\big) \otimes \mathbf{Q}\Big) \simeq \mathbf{P}^\top \mathbf{E}^\top \in \mathbb{R}^{R \times N}, \tag{26}$$

where $\mathbf{S} \in \mathbb{R}^{LN \times R^2}$.

The dynamics lie in the non-convex problem on $\{\mathbf{G}_r\}_{r \in [R]}$. A closed-form squares solution generated by $\Phi$ across the autoregression is

$$\Phi \simeq \arg\max_{\Phi} \sum_{l=1}^{L} \frac{R \sum\limits_{n=1}^{N} ||\Phi \mathbf{s}_{n,l}\mathbf{x}_{n,l} - \Phi\bar{\hat{\mathbf{s}}}^L \mathbf{x}_{n,l}||_2^2}{\sum\limits_{r=1}^{R} ||\Phi \mathbf{s}_{l,r}\mathbf{x}_l - \Phi\bar{\hat{\mathbf{s}}}^{NL}\mathbf{x}_l||_2^2}, \tag{27}$$

where $\bar{\hat{\mathbf{s}}}^L, \bar{\hat{\mathbf{s}}}^{NL}$ denotes the mean values among the dimensions. By defining the self-variate matrix $\mathbf{J}$ and the cross-variate matrix $\mathbf{K}$ for the features as

$$\mathbf{J} = \sum_{r=1}^{R} (\mathbf{s}_{l,r}\mathbf{x}_l - \bar{\hat{\mathbf{s}}}^{NL}\mathbf{x}_l)(\mathbf{s}_{l,r}\mathbf{x}_l - \bar{\hat{\mathbf{s}}}^{NL}\mathbf{x}_l)^\top,$$

$$\mathbf{K} = \sum_{n=1}^{N} (\mathbf{s}_{n,l}\mathbf{x}_{n,l} - \hat{\mathbf{s}})(\bar{\hat{\mathbf{s}}}^L \mathbf{x}_{n,l})^\top, \tag{28}$$

where the Eq. 27 is equivalent to a trace representation:

$$\mathbf{E}^\top \mathbf{G} \simeq \arg\max_\Phi \text{tr}[\Phi \mathbf{J}^{-1} \mathbf{K} \Phi^\top], \tag{29}$$

further we have one solution on the un-constrained core matrix $\mathbf{G}$ to find the correlations by the generalized eigenvalue decomposition as

$$\mathbf{G} = \frac{1}{L}^\dagger \mathbf{E} \mathbf{P}^\top \mathbf{E}^\top (\sum_{l=1}^{L} \mathbf{x}_l \mathbf{s}_l{}^\top)(\sum_{l=1}^{L} \mathbf{j}_l \mathbf{k}_l{}^\top)^{-1}. \tag{30}$$

## C  THEORETICAL VALIDATION

Here are the validations for who may wonder: (1) Decomposition: how the stationary additive decomposition could be transformed into NAILong decomposition on non-stationary time series, and (2) Froecasting: why we handle the Amplify more complex than the constraint Seasonality and Residual decomposed in DeFa.

Instead of directly applying LSTM or Transformer to capture the expression unknowingly, we are more likely to wonder why even LLM-based models fail to learn and represent simple time sereis such as $y = \exp(x)$.

### C.1  FROM ADDITION TO NAILONG: A NON-STATIONARY DECOMPOSITION TRANSFORM

*How the stationary additive decomposition could be transformed into NAILong decomposition on non-stationary time series?*

**General Definition.**  Now a general definition of non-stationary time series with one channel only from time 1 to time $N$. Given time series $\mathbf{X} = [x_1, \ldots, x_n, \ldots, x_N]$, it hold obvious trends Trend $= [t_1, \ldots, t_n, \ldots, t_N]$, $t_n > 0$ and seasonality Seasonality $= [s_1, \ldots, s_n, \ldots, s_N]$. Respectively, the residual is given as Residual $= [r_1, \ldots, r_n, \ldots, r_N]$:

$$\mathbf{X} = \text{Trend} + \text{Seasonality} + \text{Residual}, \tag{31}$$

and at time $n$, the scalar value is:

$$x_n = t_n + s_n + r_n, \tag{32}$$

We recall the definition on NAILong (Eq. 5):

$$\mathbf{Amp}_h, \mathbf{NS}_h, \mathbf{R}_h = \text{NAILong}(\mathbf{X}_h)$$
$$\hat{\mathbf{X}}_h = \mathbf{Amp}_h \odot \mathbf{NS}_h \oplus \mathbf{R}_h \tag{33}$$

**Assumptions of Non-Stationary Decomposition.**  If Seasonality further presents a pure periodic character, the challenges remaining in non-stationary data exist as the statistics characters would change with time. Here we denote $\mathbf{A} = [a_1, \ldots, a_n, \ldots, a_N]$, $a_n > 0$, $\mathbf{B} = [b_1, \ldots, b_n, \ldots, b_N]$ and $C = [c_1, \ldots, c_n, \ldots, c_N]$ as parameters changed among time on the simple trends or seasonalities or the residual, and $\odot$ as the Hadamard product:

$$\mathbf{X} = \mathbf{A} \odot \text{Trend} + \mathbf{B} \odot \text{Seasonality} + \mathbf{C} \odot \text{Residual}, \tag{34}$$

and at time $n$, the scalar value is:

$$x_n = a_n \cdot t_n + b_n \cdot s_n + c_n \cdot r_n. \tag{35}$$

It is clear that this is why linear- or Transformer-based methods could not effectively extract the seasonality following the STL addition decomposition, where the main components vary on different looking-back windows. The reason is the value at time $n$ is effected by previous time $n - 1$ as temporal cor-relationship.

In the following paragraphs, analysing on trend, seasonality (in conditions) and frequency-domain transform will be demonstrated. We will prove how the previous addition formulation could be transformed into something like

$$x(n) = d(n) \cdot x(n-1) + y(n) \cdot \text{ns}(n), \tag{36}$$

and

$$\mathbf{X} = \mathbf{Amp} \odot \mathbf{NS} + \mathbf{R}. \tag{37}$$

**Trend Transform.** As the trend is defined as a steady function on time steps for real values (ARIMA), we could have

$$
\begin{aligned}
a_n \cdot t_n &= a(n) \cdot t(n) \\
&= a(n-1) \cdot \partial t(n-1) + \partial a(n-1) \cdot t(n-1) + \epsilon(n) + a(n-1) \cdot t(n-1) \\
&= \left(1 + \frac{\partial a(n-1)}{a(n-1)} + \frac{\partial t(n-1)}{t(n-1)}\right) \cdot a(n-1) \cdot t(n-1) + \epsilon(n-1) \\
&\approx t(0) \prod_{i=1}^{n}(1 + \partial \ln |a(i-1)| + \partial \ln |t(i-1)|) \\
&= t(0) \prod_{i=0}^{n} d(i), \text{where } d(i) = 1 + \partial \ln |a(i-1)| + \partial \ln |t(i-1)|,
\end{aligned}
\tag{38}
$$

where we define $d(i) > 0$ for any $i > 0$ (to define the whole trend positive among time). Similarly, we denote $d(i)$ as time series $D = [d_1, \ldots, d_n, \ldots, d_N]$. If we consider $\epsilon(n)$ as a complex expression on Trend:

$$
a_n \cdot t_n = t(0) \prod_{i=0}^{n} d(i) + \epsilon(n-1)
\tag{39}
$$

**Seasonality Transform.** As the seasonality part, for such discrete time series, we define the time length $N = m \cdot K + n_0$ where $K, n_0 > 0$. $K$ is a default period length. And we denote additional $P = [p_1, \ldots, p_n, \ldots, p_N]$ and $Q = [q_1, \ldots, q_n, \ldots, q_N]$ as time functions $p(n)$ and $q(n)$ for time steps. Ideally for a seasonality, $p(n_1) = p(n_2)$ and $q(n_1) = q(n_2)$ could be satisfied for any time steps $n_1$ and $n_2$. So we could extract a normalized seasonality ns from $b(n) \cdot s(n) = p(n) \cdot \text{ns}(n) + q(n)$ of a non-stationary seasonality behaving, then we sum up every seasonality value in $\mathbf{B} \odot$ Seasonality (we mark the sum ends with different time step). The seasonality value $\mathbf{B} \odot$ Seasonality at time $t$ is

$$
b(n) \cdot s(n) = b(n) \cdot p(n) \cdot \text{ns}(n) + b(n) \cdot q(n)
\tag{40}
$$

So the non-stationary definition in Eq. 35 could be

$$
x(n) = t(0) \prod_{i=0}^{n} d(i) + \hat{b}(n) \cdot \text{ns}(n) + \hat{r}(n),
\tag{41}
$$

where $\hat{b}(n)$ and $\hat{r}(n)$ satisfy

$$
\hat{b}(n) = b(n) \cdot p(n), \ \hat{r}(n) = b(n) \cdot q(n) + c(n) \cdot r(n).
\tag{42}
$$

Now we could have a general and non-stationary formulation of the addition decomposition:

$$
x(n) = d(n) \cdot x(n-1) + \hat{b}(n) \cdot \text{ns}(n) - d(n) \cdot \hat{b}(n-1) \cdot \text{ns}(n-1) + \hat{r}(n) - \hat{r}(n-1) \cdot d(n), \tag{43}
$$

where the trend and seasonality expressions are slightly different with previous ARIMA-type non-stationary series considering the time relationship with all previous time steps. This is prepared for later adaptive interactions. Recent methods try frequency-domain features or patch features learning to adapt such complex time-dependent trend or seasonality. Not difficult to discover that

$$
\boxed{\hat{x}(n) = d(n) \cdot \hat{x}(n-1) + y(n) \cdot \text{ns}(n)},
\tag{44}
$$

where

$$
\begin{aligned}
\hat{x}(n) &= x(n) - \hat{r}(n), \\
y(n) &\approx \hat{b}(n) - \hat{b}(n - K + 1).
\end{aligned}
\tag{45}
$$

To prove Eq. 44, we first denote $\hat{s}(n)$ as

$$
\begin{aligned}
\hat{s}(n) &= \hat{b}(n) \cdot \mathrm{ns}(n) - d(n) \cdot \hat{b}(n-1) \cdot \mathrm{ns}(n-1) \\
&= \hat{b}(n) \cdot \mathrm{ns}(n) - d(n) \cdot [\hat{s}(n-1) + d(n-1) \cdot \hat{b}(n-2) \cdot \mathrm{ns}(n-2)] \\
&= \hat{b}(n) \cdot \mathrm{ns}(n) - d(n) \cdot d(n-1) \cdot \hat{b}(n-2) \cdot \mathrm{ns}(n-2) - d(n) \cdot \hat{s}(n-1) \\
&\quad \cdots \\
&= \hat{b}(n) \cdot \mathrm{ns}(n) - \hat{s}(1) \prod_{i=1}^{n} d(i) - \sum_{i=1}^{n-1} \hat{s}(i) \prod_{j=i}^{n-1} d(i+1)
\end{aligned}
\tag{46}
$$

then for time step pairs $(n, n+1)$ we have

$$
\begin{aligned}
\hat{s}(n+1) - \hat{s}(n) = {}& \hat{b}(n+1) \cdot \mathrm{ns}(n+1) - \hat{b}(n) \cdot \mathrm{ns}(n) \\
& - \left( (d(n+1)-1) \cdot \underbrace{\left( \sum_{i=1}^{n-1} \left( \hat{s}(i) \prod_{j=i}^{n-1} d(i+1) \right) + \hat{s}(1) \prod_{i=1}^{n} d(i) + \hat{s}(n) \right)}_{\text{denoted as } f(n)} + \hat{s}(n) \right),
\end{aligned}
\tag{47}
$$

then we have

$$
\hat{s}(n+1) = \hat{b}(n+1) \cdot \mathrm{ns}(n+1) - \hat{b}(n) \cdot \mathrm{ns}(n) - (d(n+1)-1) \cdot f(n).
\tag{48}
$$

And for the second time step pair $(n, n+K)$ where $\mathrm{ns}(n+K) = \mathrm{ns}(n)$, we consider

$$
\begin{aligned}
\hat{s}(n+K) - \hat{s}(n) = {}& (\hat{b}(n+K) - \hat{b}(n)) \cdot \mathrm{ns}(n+K) \\
& - \left( (\prod_{i=1}^{K} d(n+i) - 1) \cdot f(n+K-1) + \hat{s}(n) \right),
\end{aligned}
\tag{49}
$$

then we have

$$
\hat{s}(n+K) = (\hat{b}(n+K) - \hat{b}(n)) \cdot \mathrm{ns}(n+K) - (\prod_{i=1}^{K} d(n+i) - 1) \cdot f(n+K-1).
\tag{50}
$$

When we extend the time step from $n+1$ to $n+K$ within one period based on Eq. 48, we have

$$
\hat{s}(n+K) = \hat{b}(n+K) \cdot \mathrm{ns}(n+K) - \hat{b}(n+K-1) \cdot \mathrm{ns}(n+K-1) - (d(n+K)-1) \cdot f(n+K-1),
\tag{51}
$$

then compared to Eq. 49 while we hold $d(i) > 1$ from Eq. 38:

$$
\begin{aligned}
\hat{s}(n+K) = {}& \hat{b}(n+K) \cdot \mathrm{ns}(n+K) \\
& \underbrace{- \hat{b}(n+K-1) \cdot \mathrm{ns}(n+K-1) - (d(n+K)-1) \cdot f(n+K-1)}_{n+1 \text{ special}} \\
= {}& \hat{b}(n+K) \cdot \mathrm{ns}(n+K) \\
& \underbrace{- \hat{b}(n) \cdot \mathrm{ns}(n+K) - (\prod_{i=1}^{K} d(n+i) - 1) \cdot f(n+K-1)}_{n+K \text{ special}}.
\end{aligned}
\tag{52}
$$

**Seasonality Transform Condition 1: If** $d(n+K) < 1$. Then $\prod_{i=1}^{K} d(n+i) - d(n+K) < \epsilon$ where $\epsilon$ is a small positive value, we have

$$
\hat{b}(n+K-1) \cdot \mathrm{ns}(n+K-1) - \hat{b}(n) \cdot \mathrm{ns}(n+K) < \epsilon,
\tag{53}
$$

but actually as a normalized seasonality, it could be satisfied only when $\hat{b}(n+K-1) \cdot \mathrm{ns}(n+K-1) = \hat{b}(n) \cdot \mathrm{ns}(n+K)$ is possible, and this means $\hat{s}(n+K) = (\hat{b}(n+K) - \hat{b}(n)) \cdot \mathrm{ns}(n+K)$. This somehow reveals the fact that the forecasting would occur the imbalance between point-wise (Linear) and patch-wise (Transformer) predictions, since we have no idea how trends would behave.

**Seasonality Transform Condition 2: If** $d(n + K) > 1$. Then $d(n + K) \gg \prod_{i=1}^{K} d(n + i) - d(n + K) > 1$. We move further for both the short- and long-term interactions when $K > 1$ ($K = 1$ is an obvious one where a point is the smallest period) to dig in:

$$
\big(\prod_{i=1}^{K} d(n + i) - d(n + K)\big) \cdot f(n + K - 1) = \hat{b}(n + K - 1) \cdot \text{ns}(n + K - 1) - \hat{b}(n) \cdot \text{ns}(n)
$$

$$
\Rightarrow f(n + K - 1) = \frac{\hat{b}(n + K - 1) \cdot \text{ns}(n + K - 1) - \hat{b}(n) \cdot \text{ns}(n)}{\prod_{i=1}^{K} d(n + i) - d(n + K)}
$$

$$
\Rightarrow \underbrace{\sum_{i=1}^{n+K-2} \big(\hat{s}(i) \prod_{j=i}^{n+K-2} d(i+1)\big) + \hat{s}(1) \prod_{i=1}^{n+K-1} d(i) + \hat{s}(n + K)}_{\text{left}(n+K-1)}
$$

$$
= \underbrace{\frac{\hat{b}(n + K - 1) \cdot \text{ns}(n + K - 1) - \hat{b}(n) \cdot \text{ns}(n)}{\prod_{i=1}^{K} d(n + i) - d(n + K)}}_{\text{right}(n+K-1)},
$$

$$(54)$$

where we denote the left side as $\text{left}(n + K - 1)$ and the right side as $\text{right}(n + K - 1)$.

Now we denote a sub-part of $\text{left}(n + K - 1)$ as

$$
\text{spr}(n + K - 1) = \sum_{i=1}^{n+K-2} \big(\hat{s}(i) \prod_{j=i}^{n+K-2} d(i+1)\big) + \hat{s}(1) \prod_{i=1}^{n+K-1} d(i), \tag{55}
$$

where $\text{left}(n+K-1) = \text{sp}(n+K-1) + \hat{s}(n+K)$. Then $\text{right}(n+K-1) = \text{sp}(n+K-1) + \hat{s}(n+K)$. Furthermore in $\text{left}(n + K - 1)$ we have

$$
\text{spr}(n + K - 1) = \sum_{i=1}^{n+K-2} \big(\hat{s}(i) \prod_{j=i}^{n+K-3} d(i+1)\big) + \hat{s}(n + K - 2) \prod_{j=n+K-2}^{n+K-2} d(j - 1 + 1)
$$

$$
+ \hat{s}(1) d(n + K - 1) \prod_{i=1}^{n+K-2} d(i)
$$

$$
= \sum_{i=1}^{n+K-3} \big(\hat{s}(i) d(n + K - 2) \prod_{j=i}^{n-2} d(i+1)\big)
$$

$$
+ \hat{s}(n + K - 3) d(n + K - 2) + \hat{s}(1) d(n + K - 1) \prod_{i=1}^{n+K-2} d(i)
$$

$$
= d(n + K - 1) \big( \sum_{i=1}^{n-2} \big(\hat{s}(i) \prod_{j=i}^{n-2} d(i+1)\big) + \hat{s}(1) \prod_{l=1}^{n-1} d(l)\big)
$$

$$
+ \hat{s}(n + K - 2) d(+K - 1)
$$

$$
= d(n + K - 1) \cdot \text{spr}(n + K - 2) + \hat{s}(n + K - 2) \cdot d(n + K - 1),
$$

$$(56)$$

then

$$
\text{left}(n + K - 1) = d(n + K - 1) \cdot \text{sp}(n + K - 2) + \hat{s}(n + K - 2) \cdot d(n + K - 1) + \hat{s}(n + K). \tag{57}
$$

So for $\hat{s}(n + K)$, we could have

$$
\hat{s}(n + K) = \text{left}(n + K - 1) - d(n + K - 1) \cdot \text{left}(n + K - 2), \tag{58}
$$

and $\hat{s}(n)$ presented in right:

$$
\hat{s}(n + K) = \text{right}(n + K - 1) - d(n + K - 1) \cdot \text{right}(n + K - 2). \tag{59}
$$

We recall the $\text{right}(n + K - 2)$ from $\text{right}(n + K - 1)$ in Eq. 54:

$$\text{right}(n + K - 2) = \frac{\big(\hat{b}(n + K - 2) - \hat{b}(n - 1)\big) \cdot \text{ns}(n - 1)}{\prod_{i=1}^{K} d(n + i - 1) - d(n + K - 1)}, \tag{60}$$

now we substitute Eq. 59:

$$\hat{s}(n + K - 1) = \frac{\text{ns}(n)(\hat{b}(n + K - 1) - \hat{b}(n))}{\prod_{i=1}^{K} d(n + i) - d(n + K)} - d(n + K - 1) \frac{\text{ns}(n - 1)(\hat{b}(n + K - 2) - \hat{b}(n - 1))}{\prod_{i=1}^{K} d(n + i - 1) - d(n + K - 1)}, \tag{61}$$

although there is nothing close to the approximation $y(n)$ in Eq. 45. We continue to discover how to get the simple formulation Eq. 44. We denote

$$\text{dp}(n + K) = \prod_{i=1}^{K-1} d(n + i) - d(n + K) > 0, \tag{62}$$

then we look back to the $\hat{s}(n)$ denotation in Eq. 46:

$$\begin{aligned}
\hat{s}(n + K) &= \frac{\text{ns}(n)\big(\hat{b}(n + K - 1) - \hat{b}(n)\big)}{\text{dp}(n)} - d(n + K - 1) \frac{\text{ns}(n - 1)\big(\hat{b}(n + K - 2) - \hat{b}(n - 1)\big)}{\text{dp}(n - 1)} \\
&= \frac{\big(\hat{b}(n + K - 1) - \hat{b}(n)\big)}{\text{dp}(n + K)} \text{ns}(n + K) \\
&\quad - d(n + K - 1) \frac{\big(\hat{b}(n + K - 2) - \hat{b}(n - 1)\big)}{\text{dp}(n + K - 1)} \text{ns}(n + K - 1).
\end{aligned} \tag{63}$$

If $\text{dp}(n + K) \gg 1$, it implies the trend $d(n)$ holds much bigger values than the seasonality part $\text{ns}(n)$ so the seasonality part could be treat as a trend part as well. Then we could re-write Eq. 44 and Eq. 45:

$$\hat{x}(n) = d(n) \cdot \hat{x}(n - 1), \tag{64}$$

where

$$\hat{x}(n) \approx x(n) - \hat{r}(n) + \frac{\hat{b}(n + K - 1) - \hat{b}(n)}{\text{dp}(n)} \text{ns}(n). \tag{65}$$

Else $\text{dp}(n + K) > 1$ would be converged to 1, then $\text{dp}(n + K - 1) \gg d(n + K - 1)$, so

$$\hat{s}(n + K) \approx \text{ns}(n)\big(\hat{b}(n + K - 1) - \hat{b}(n)\big). \tag{66}$$

Finally we get the formulation in Eq.44:

$$\boxed{\hat{x}(n) = d(n) \cdot \hat{x}(n - 1) + y(n) \cdot \text{ns}(n)}, \tag{67}$$

where we put seasonality into $\hat{r}(n)$ when $\text{dp}(n + K) \gg 1$

$$\begin{aligned}
\hat{x}(n) &= x(n) - \hat{r}(n), \\
y(n) &\approx \hat{b}(n) - \hat{b}(n - K + 1).
\end{aligned} \tag{68}$$

One may notice sometimes we denote the time step as $n + K$ and sometimes just $n$ where we define a period length $K$ before. When $0 < n < K$, the value at $n - K$ from inputs is 0 as default. And this is actually a padding technique when we load the data in Pytorch.

**Frequency Domain Transform to NAILong.** Now we look back to Eq. 44. In the frequency domain, discrete time series would be formulated as a convolution $*$:

$$\begin{aligned}
\hat{X}(z) &= D(z)z^{-1}\hat{X}(z) + [Y(z) * NS(z)] \\
&\approx \frac{[Y(z) * NS(z)]}{1 - D(z)z^{-1}}.
\end{aligned} \tag{69}$$

With previous $d(i) > 0$, we have $(1 - D(z)z^{-1}) > 0$ to find a reverse convolution $\hat{D}^{-1}(z)$ exists:

$$\hat{X}(z) = (\hat{D}^{-1}(z) \cdot Y(z)) * NS(z), \tag{70}$$

then back in the time domain, we would get the multiplication interaction

$$\hat{x}(n) = \hat{d}(n) \cdot ns(n), \tag{71}$$

and

$$x(n) = \hat{d}(n) \cdot ns(n) + \hat{r}(n). \tag{72}$$

**NAILong.** And this is the definition of the proposed NAILong decomposition in Sec. 3:

$$\hat{\mathbf{X}} = \mathbf{Amp} \odot \mathbf{NS} + \mathbf{R}, \tag{73}$$

represented by decomposed Amplifier ($\mathbf{Amp}$), normalized Seasonality ($\mathbf{NS}$), or Residuals ($\mathbf{R}$). Notably, the real Amplifier $\mathbf{Amp}$ is an unknown combination of $\hat{d}(n)$ and $\epsilon(n-1)$ from Eq. 38.

**NAILong Estimation.**

$$\min_{\mathbf{Amp},\mathbf{NS},\mathbf{R}} \frac{1}{2}||\mathbf{X} - \mathbf{Amp} \odot \mathbf{NS} - \mathbf{R}||_2^2, \tag{74}$$

would converge to 0 if and only if $d(i) > 0$ for any i¿0. And $d(i) = 0$ implies $t(i) = 0$. This is why we define the Amplifier as a non-negative sequence. In details, we need to address NAILong's optimization as it is a simple yet efficient decomposition. An alternating minimization algorithm is applied in which we have a set of sub-optimizations on $\mathbf{Amp}, \mathbf{NS}, \mathbf{R}$ respectively. And these are just simple convex optimizations to each where the remaining ones are fixed.

## C.2 Specialized Amplifier Forecasting: Constraints from Normed Seasonality and Sparse Residual

*Why we handle the Amplify more complex than the Seasonality and Residual?*

**Normalized Seasonality.** One may wonder if the interaction of the amplifier and the seasonality is still wildly out of control, where the adaptive interaction we design would shift at a time $t$ that $2i < N$:

$$\underbrace{\mathrm{unit}(i)}_{\mathbf{Amp}} \cdot \underbrace{\sin(2i)}_{\mathrm{ns}} = \underbrace{2\cos(i)}_{\mathbf{Amp}} \cdot \underbrace{\sin(i)}_{\mathrm{ns}}. \tag{75}$$

To ensure the factorization admits a unique solution, we impose a set of formal constraints on the seasonal component. Specifically, we define the seasonality $\mathbf{NS}$ such that $\max_n |ns(n)| = 1$, ensuring that all signal magnitude is attributed solely to the amplifier component $\mathbf{Amp}$. Moreover, the amplifier is expressed exclusively as holding the high-frequency parts to restrict $\mathbf{NS}$ to lie within a low-dimensional, band-limited subspace. We address the potential presence of high-frequency parts in $\mathbf{Amp}$ in the next subsection.

**Sparse Residual in Freuqnecy-domain.** The reason why the specialized amplifier forecasting is a bit more complex than a linear layer is we want to make the spatialize of the residual. And the periodic part of the residual would complex the representation of the Amplifier $\hat{d}(n) + \epsilon(n-1)$. The residual transformation in the frequency domain from $\hat{R}(z)$ to $\hat{R}'(z)$ is extended here, where we want to simplify its representation to a linear layer learning formulation. $\hat{R}'(z)$ denotes a sparse residual correction term, derived from the original residual $\hat{R}(z)$.

The original formulation in Eq. 69 is:

$$X(z) = \underbrace{\frac{Y(z) \cdot NS(z)}{1 - D(z)z^{-1}}}_{\text{Primary Term}} + \underbrace{\frac{\hat{R}(z)}{1 - D(z)z^{-1}}}_{\text{Residual Term}} = \text{Primary} + \hat{R}'(z), \tag{76}$$

where $\hat{R}'(z)$ is the result of applying a sparse operator to reduce the frequency or temporal complexity of the correction term. While the residual may preserve critical dynamic events (e.g., bursts,

anomalies) that are not captured by the main system model, it implies the residual may remain orthogonal or complementary to the primary term in high-frequency or transient regions.

This sparsity approximation problem could be formulated as one L1 regularization optimization:

$$\min_{\mathcal{T}_\eta} \|\hat{R}(z) - \mathcal{Z}\{\mathcal{T}_\eta(\hat{r}[n])\}\|_1 + \lambda\|\mathcal{Z}\{\mathcal{T}_\eta(\hat{r}[n])\}\|, \tag{77}$$

where we define such a time series mask $\eta$ to suppress insignificant residual values in the time domain:

$$\hat{r}'[n] = \mathcal{T}_\eta(\hat{r}[n]) = \begin{cases} \hat{r}[n], & |\hat{r}[n]\| > \eta \\ 0, & \text{otherwise} \end{cases} \quad \Rightarrow \quad \hat{R}'(z) = \mathcal{Z}\{\hat{r}'[n]\}. \tag{78}$$

Since the sparse residual is denoted as high-frequency information, and we have $(1 - D(z)z^{-1}) > 0$, the information of periodic events $\hat{R}(z)$ that may argued to conduct aside from the seasonality could be re-formulated as:

$$\begin{aligned} \hat{R}'(z) &= \frac{\hat{R}(z)}{1 - D(z)z^{-1}} = \hat{R}(z)(1 + D(z)z^{-1} + D^2(z)z^{-2} + \cdots) \\ &= \hat{R}(z)(1 + D(z)z^{-1}) + \epsilon(z), \end{aligned} \tag{79}$$

then for the primary could be achieved where $\|D(z)z^{-1}\| < 0$ (from $(1 - D(z)z^{-1}) > 0$)

$$\text{Primary Term} \approx Y(z) * NS(z)(1 + D(z)z^{-1}) = (\hat{D}^{-1}(z) \cdot Y(z)) * NS(z). \tag{80}$$

The issue left here is the approximation lead by the high-frequency part $\epsilon(z)$ of the residual $\hat{R}(z)$. Based on the time series mask we defined earlier, it would be a formulation based on the gap of $\mathcal{T}_\eta$ at N related to the length of the time series:

$$\min_{\mathcal{T}_\eta, N} \|\hat{R}(z)(1 + D(z)z^{-1}) - \mathcal{Z}\{\mathcal{T}_\eta(\hat{r}[i])\}_{i=0}^N + \epsilon(z) - \mathcal{Z}\{\mathcal{T}_\eta(\hat{r}[i])\}_{N+1}^n\|_1 + \lambda N \min_N \sum_{i=N+1}^n |\hat{r}[i]|, \tag{81}$$

and this could be optimized by learning the $\mathcal{T}_\eta$ as a linear layer where $\mathcal{T}_\eta$ is originally defined as a mask in the time domain.

Now if we go back to the representation of the Amplifier $\mathbf{A} = \hat{d}(t)$, the complex part would be

$$\hat{D}^{-1}(z) = (1 + D(z)z^{-1}) + \frac{\hat{R}(z)}{Y(z)} \approx \underbrace{1 + D(z)z^{-1}}_{\text{smooth part}}, \tag{82}$$

and that is why we design a more complex network than one linear layer to forecast the Amplifier to reduce the smooth lines in the predictions.

