# OpenReview forum: "DeFa: Non-Stationary Decomposition and Factorized Forecasting for Multivariate Time Series"
_ICLR.cc/2026/Conference — Submitted to ICLR 2026_

### Official Review · Reviewer_KLhB · 2025-10-19

**Soundness:** 2
**Presentation:** 2
**Contribution:** 2
**Rating:** 2
**Confidence:** 4

**Summary:**

This paper proposes DeFa, a framework for non-stationary multivariate time series forecasting. DeFa consists of two main components: (1) NAILong, a decomposition strategy that separates a time series into a time-varying Amplifier, normalized Seasonality, and sparse Residual via a multiplicative formulation ($X = Amp · NS + R$), and (2) FaTA, a factorized tensor autoregression module designed to forecast the complex dynamics of the Amplifier component. The authors evaluate DeFa on several real-world benchmarks, claiming state-of-the-art performance, and demonstrate its utility as a plug-in module for existing models.

**Strengths:**

$\textbf{Ambitious Motivation}$: The paper tackles the important and challenging problem of modeling non-stationary dynamics in multivariate time series, which is a relevant and timely research direction.

$\textbf{Novel Formulation}$: The proposed NAILong decomposition moves beyond simple additive models to a multiplicative-interaction-based formulation ($Amp · NS + R$). This is a conceptually interesting approach for capturing time-varying patterns.

$\textbf{Comprehensive Experiments}$: The paper provides extensive experimental results across seven standard datasets and multiple forecasting horizons, comparing against a wide array of modern baselines from different families (linear, Transformer, frequency-based).

**Weaknesses:**

1. The core methodological contributions—NAILong and FaTA—are described with overwhelming and often confusing complexity. Key components like the Multi-Component Adaptive Filter (M-CAF) and Interactive Component Gate (ICG) are explained with a barrage of operations (e.g., "multi-resolution reshaping," "sparse circular convolutions") without providing an intuitive, step-by-step understanding of how the input X is concretely transformed into Amp, NS, and R.

2. The model incorporates numerous complex techniques (e.g., randomized orthogonal projections for permutation invariance, sparse convolutions with dynamically predicted filters). The paper fails to provide clear, principled justifications for these specific choices. The architecture appears as a collection of sophisticated components rather than a coherent, well-motivated design, raising concerns about whether its performance stems from genuine innovation or extensive engineering.

3. The ablation study (Table 4) is inadequate for validating the design: 1) Variants (A) and (B) test data efficiency, not architectural components; 2) The catastrophic failure of the additive baseline (Variant D) is a red flag. A simple additive model (Amp + NS + R) should not perform orders of magnitude worse than the multiplicative one (Amp · NS + R) if both are implemented fairly. This suggests a potential implementation flaw or an unfairly handicapped baseline, rather than conclusive evidence for the multiplicative formulation; 3) There is no ablation on the core, complex components of FaTA (e.g., the sparse convolution, the orthogonal projection).

4. The paper repeatedly claims the model is "interpretable" but provides zero visualization or concrete analysis of the learned components (Amp, NS, R) or factor matrices (P, Q, T). Without demonstrating what these components represent in a real-world dataset, the claim of interpretability is hollow.

**Questions:**

1. Could you provide a clear, step-by-step algorithmic pseudo-code for the entire NAILong decomposition process? The current description, especially of M-CAF, is too opaque to follow.

2. Can you detail the exact implementation of the additive baseline (Variant D)? The massive performance degradation is suspicious and suggests this baseline may not have been a faithful or competitive implementation. How were the additive components forecasted?

3. Please show a concrete example: visualize the Amp, NS, and R components for a specific variable from the ETT dataset and explain what real-world phenomenon each one captures.

4. What is the empirical evidence that the "sparse circular convolution" in the temporal mode is superior to a standard convolution or a simple linear layer? How was the key hyperparameter τ=4 determined?

5. How does DeFa compare to a straightforward application of classical multiplicative seasonality decomposition (e.g., STL with a multiplicative model) followed by a linear forecaster? The related work does not engage with this established literature.

---

### Official Review · Reviewer_D7Aw · 2025-10-21

**Soundness:** 2
**Presentation:** 1
**Contribution:** 2
**Rating:** 2
**Confidence:** 4

**Summary:**

The paper presents **DeFa**, a decomposition-then-forecast framework for multivariate non-stationary time series. It introduces **NAILong**, a multiplicative factorization $X=\text{Amp} \cdot \text{NS} + R$ that isolates a time-varying Amplifier, normalized Seasonality, and sparse Residuals. To forecast the challenging amplifier dynamics, it further proposes **FaTA**, a factorized tensor autoregression that extends Tucker with specialized temporal (sparse circular convolutions), cross-variate (permutation-invariant), and per-variate (identity-anchored) factors, plus a plug-in option.

**Strengths:**

1. The multiplicative factorization $X=\text{Amp} \cdot \text{NS} + R$ is mildly novel and cleanly separates non-stationarity from seasonality/residuals.
2. The manuscript is detailed, signaling substantial effort and execution (extensive experiments, ablations, and analyses).

**Weaknesses:**

1. Unclear pain-point alignment: The paper doesn’t show which concrete shortcomings of prior work it solves. The abstract flags “over-parameterization and difficulty modelling shifting patterns in simple short- and long-term terms,” but the proposed modules aren’t clearly tied to these issues; the core contribution remains vague.

2. Modest empirical gains: Improvements over strong SOTA baselines are small and not consistently significant.

3. Immature writing; needs substantial polishing: prior-work discussion is disorganized and lacks a clear narrative.
    - Use `\citep` (not `\cite`) when the cited work isn’t the sentence subject; this is misused almost throughout.
    - Introduction paragraph 2 (lines 42–64): prior-work discussion is disorganized and lacks a clear narrative.
    -  Notation inconsistency: Introduction paragraph 4 (lines 71–80) starts with $Amp$, $NS$, $R$, but later uses $\textbf{Amp}$, $\textbf{NS}$, $\textbf{R}$ for the same modules, causing confusion.

**Questions:**

See **Weaknesses**.

**Details Of Ethics Concerns:**

None.

---

### Official Review · Reviewer_LEKQ · 2025-10-23

**Soundness:** 3
**Presentation:** 2
**Contribution:** 3
**Rating:** 2
**Confidence:** 3

**Summary:**

The paper proposes DeFa, a unified decomposition-and-forecasting framework designed for non-stationary multivariate time-series forecasting. outperforming the state-of-the-art methods in terms of both interpretable forecasting accuracy and scalability.

**Strengths:**

1. This paper presents a novel methodological framework on predicting non-stationary time series.
2. The tucker factorization in FaTA module provides physical interpretability and plausibility.
3. The proposed method is GPU-memory and computationally efficient.

**Weaknesses:**

1. weak empirical results: the proposed method fails to achieve SOTA in more than 1/3 of the total benchmarks.
2. This paper has to be re-organized and is hard to follow.
3. The benchmarks are quite limited to the standard ones in time series forecasting. The author may need to include more non-stationary time series dataset to prove the effectiveness of his/her proposed method.
4. A lack of visual interpretability validation.

**Questions:**

1. Is this method only limited to small dataset or time series forecasting task?
2. During the factorization process, is it possible to have multiple suboptimal solutions that may affect the effectiveness of predictions?

---

### Official Review · Reviewer_i6MB · 2025-10-28

**Soundness:** 3
**Presentation:** 3
**Contribution:** 3
**Rating:** 6
**Confidence:** 4

**Summary:**

This paper proposes DeFa, a “decompose-then-forecast” framework for long-horizon multivariate time series. First, NAILong splits the series into three components: an amplifier (Amp) that captures time-varying and cross-channel interactions, a relatively stationary seasonal component (NS), and a sparse residual (R) that absorbs anomalies, using a multiplicative coupling to match non-stationary scaling effects. Then, FaTA applies low-rank factorization to the autoregressive coefficient tensor of Amp, selects sparse key lags in the time dimension, and imposes permutation-invariance and physical interpretability constraints in the variable dimensions; NS and R are extrapolated with lightweight linear heads. Training jointly optimizes historical reconstruction + future forecasting. Experiments across strong baselines and standard datasets show stable advantages for long-horizon prediction, and DeFa can also serve as a plug-in to boost other models with noticeable gains.

**Strengths:**

S1.Replaces common additive decomposition with multiplicative coupling, which better matches the intuition of trend-driven dynamic scaling of seasonal patterns; the ICG imposes positivity/symmetry/sparsity priors and gated interactions across the three components, providing a novel structural inductive bias.

S2. Clear module boundaries and data flow; figures and formulas are consistent, which facilitates reproduction and portability.

S3. As a preprocessor/plug-in, it can broadly enhance existing models, yielding long-horizon gains across multiple baseline families with small extra overhead, strong engineering practicality.

**Weaknesses:**

W1. The relationship to prior “decomposition + lightweight extrapolation/linear head/frequency-domain” lines is under-positioned. For example, Autoformer/FEDformer also introduce decomposition or frequency modeling; CoST/TimeDRL explore decomposable or disentangled representations. Provide a more systematic contrast, both theoretically and empirically, highlighting essential differences.

W2. Sparse lag convolution and permutation-invariant projection are key designs, but current support relies mainly on error metrics and qualitative intuition. Add stability analyses of lag selection (consistency of nonzero lags in T under different initializations/random seeds) and permutation perturbation tests (error changes and factor similarity before/after shuffling channel order) to substantiate the claims of physicality/permutation invariance.

W3. Achieving “permutation insensitivity” via random orthogonal projection does not strictly guarantee true invariance: the inverse projection and parameter sharing downstream may still be sensitive to index order; projection can also introduce information loss and extra variance.

**Questions:**

Q1. When the trend is negative or highly non-monotonic, do the nonnegativity and smoothing assumptions on Amp cause underfitting? Have you considered a signed amplifier or a dual-channel Amp± alternative?

Q2. How stable is the selection of sparse lags?

---

### Meta-Review · Area_Chair_3Bir · 2026-01-07

**Summary:**

This paper presents a framework for non-stationary multivariate time series forecasting.

Reviewers generally agree that the paper addresses an important and challenging problem, namely, long-horizon forecasting under non-stationary multivariate settings. The proposed direction is considered relevant and timely. However, a central concern shared across reviewers is that the paper does not clearly articulate which concrete limitations of prior methods DeFa fundamentally resolves, nor does it sufficiently position its contributions relative to existing decomposition-based and representation-learning approaches. In addition, multiple reviewers note that the proposed framework introduces a large number of sophisticated components (e.g., M-CAF, ICG, sparse circular convolutions, orthogonal projections, and Tucker factorization), but the paper does not convincingly demonstrate which specific design choices are essential for the observed performance gains, nor what fundamental limitation of prior work is being resolved.


The paper received three scores recommending rejection and one score marginally above the acceptance threshold. While the positive review acknowledges the potential of the proposed direction, the negative reviews consistently identify unresolved technical and empirical issues that are considered blocking. These concerns were not resolved during the rebuttal phase. Given that the main concerns, i.e., particularly regarding unclear core contributions and weak positioning with respect to prior work, have not been adequately addressed, I recommend that the paper not be accepted in its current form. The area chairs encourage the authors to carefully address the above issues in a revised version, with clearer articulation of the core insight, stronger justification of key design choices, and improved positioning relative to existing literature.

**Reviewer Concerns:**

The authors did not respond to the reviewers' concerns. Consequently, the technical and empirical concerns raised by the reviewers remain unresolved.

**Reviewer Scores:**

The authors did not respond to the reviewers' concerns.
All the reviewers would like to maintain their original scores.

---

### Decision · Program_Chairs · 2026-01-26

Reject